# Structural relationship between the putative hair cell mechanotransduction channel TMC1 and TMEM16 proteins

Angela Ballesteros[1], Cristina Fenollar-Ferrer[2,3,4], Kenton Jon Swartz[1]*

[1]Molecular Physiology and Biophysics Section, National Institute of Neurological Disorders and Stroke, National Institutes of Health, Bethesda, United States; [2]Laboratory of Molecular & Cellular Neurobiology, National Institute of Mental Health, National Institutes of Health, Bethesda, United States; [3]Laboratory of Molecular Genetics, National Institute of Deafness and Other Communication Disorders, National Institutes of Health, Bethesda, United States; [4]Molecular Biology and Genetics Section, National Institute of Deafness and Other Communication Disorders, National Institutes of Health, Bethesda, United States

**Abstract** The hair cell mechanotransduction (MET) channel complex is essential for hearing, yet it's molecular identity and structure remain elusive. The transmembrane channel–like 1 (TMC1) protein localizes to the site of the MET channel, interacts with the tip-link responsible for mechanical gating, and genetic alterations in TMC1 alter MET channel properties and cause deafness, supporting the hypothesis that TMC1 forms the MET channel. We generated a model of TMC1 based on X-ray and cryo-EM structures of TMEM16 proteins, revealing the presence of a large cavity near the protein-lipid interface that also harbors the Beethoven mutation, suggesting that it could function as a permeation pathway. We also find that hair cells are permeable to 3 kDa dextrans, and that dextran permeation requires TMC1/2 proteins and functional MET channels, supporting the presence of a large permeation pathway and the hypothesis that TMC1 is a pore forming subunit of the MET channel complex.
DOI: https://doi.org/10.7554/eLife.38433.001

*For correspondence:
swartzk@ninds.nih.gov

## Introduction

The mechanotransduction (MET) channel in hair cells of the vertebrate inner ear convert the mechanical stimulus produced by sound waves into electrical signals that are transmitted into the central nervous system and is therefore essential for the perception of sound (*Fettiplace and Kim, 2014*; *Gillespie and Müller, 2009*). Hair cells are highly specialized sensory cells capable of detecting small and rapid displacements of the mechanosensitive stereocilia bundle at the cell apex (*Corey and Hudspeth, 1983*). The sensory stereocilia bundle consists of three rows of actin-based stereocilia arranged in a staircase fashion and connected by an extracellular protein filament or tip link that is formed by protocadherin 15 (PCDH15) and cadherin 23, and that mechanically gates the MET channel (*Ahmed et al., 2006*; *Fettiplace and Kim, 2014*; *Furness et al., 2008*; *Gillespie and Müller, 2009*; *Kachar et al., 2000*; *Kazmierczak et al., 2007*; *Sotomayor et al., 2012*).

Electrophysiological studies of hair cells reveal that the MET channel is a large conductance (150–300 pS in low $Ca^{2+}$) and cation-selective channel (*Beurg et al., 2006*; *Corey and Hudspeth, 1979*; *GéléocGeleoc et al., 1997*; *Kros et al., 1992*; *Ohmori, 1985*) that is permeable to relatively large organic compounds, including quaternary ammonium ions, FM1-43 and aminoglycoside antibiotics (*Alharazneh et al., 2011*; *Gale et al., 2001*; *Marcotti et al., 2005*; *Meyers et al., 2003*; *Ohmori, 1985*). Although the molecular identity and structure of the MET channel remains elusive, a

growing body of evidence supports the hypothesis that the transmembrane channel-like 1 and 2 (TMC1 and TMC2) proteins are pore-forming subunits of the MET channel. TMC1 is the most abundant subtype expressed in mature auditory hair cells, whereas TMC2 is more abundantly expressed early in postnatal development, coinciding with the onset of mechanotransduction (*Corns et al., 2017*; *Kawashima et al., 2011*, *2015*; *Kim and Fettiplace, 2013*). Hair cells from TMC1 and TMC2 double knockout mice appear structurally unperturbed at P5, yet exhibit no MET channel currents (*Pan et al., 2013*). Both TMC1 and TMC2 have been shown to biochemically interact with the tip link that is required for mechanical gating of the MET channel (*Assad et al., 1991*; *Maeda et al., 2014*). In addition, both TMC1 and TMC2 are expressed at the tip of the shorter stereocilia (*Kurima et al., 2015*) where the MET channel has been localized (*Beurg et al., 2009*). Finally, the deafness-causing TMC1 Beethoven (*Bth*) mutation (M418K in hTMC1 or M412K in mTMC1) alters the $Ca^{2+}$ permeability, single channel conductance and dihydrostreptomycin blockade of the MET channel (*Beurg et al., 2015a*; *Corns et al., 2016*; *Pan et al., 2013*). However, attempts to reconstitute channel activity with TMC proteins have not been successful to date because they are retained in the endoplasmic reticulum and do not traffic to the plasma membrane of mammalian cells (*Guo et al., 2016*; *Kawashima et al., 2011*; *Labay et al., 2010*; *Maeda et al., 2014*). In addition, genetic and biochemical evidence suggest that TMHS (tetraspan membrane protein of hair cell stereocilia) (*Beurg et al., 2015b*; *Xiong et al., 2012*), TMIE (transmembrane inner ear expressed protein) (*Park et al., 2013*; *Zhao et al., 2014*) and CIB2 ($Ca^{2+}$ and integrin binding family member 2) (*Giese et al., 2017*) may contribute to forming the MET channel complex.

The TMC proteins have been proposed to be related to TMEM16 proteins (*Hahn et al., 2009*; *Medrano-Soto et al., 2018*), a fascinating family of membrane proteins that have been shown to function as $Ca^{2+}$-activated $Cl^-$ channels (TMEM16A,B) (*Dang et al., 2017*; *Hartzell and Whitlock, 2016*; *Paulino et al., 2017*; *Peters et al., 2018*; *Schroeder et al., 2008*; *Yang et al., 2008*) or as $Ca^{2+}$-activated lipid scramblases (e.g. TMEM16F) (*Brunner et al., 2014*; *Malvezzi et al., 2013*; *Yang et al., 2012*). In the present study, we evaluated the relationship between TMC1 and TMEM16 proteins, and used the X-ray structure of the *Nectria hematococca* TMEM16 (nhTMEM16) (*Brunner et al., 2014*) phospholipid scramblase and the cryo-electron microscopy (cryo-EM) structures of the mouse TMEM16A (mTMEM16A) $Ca^{2+}$-activated $Cl^-$ channel (*Paulino et al., 2017*) (also see [*Dang et al., 2017*]) as templates to model the structure of mouse TMC1 (mTMC1). Our mTMC1 models establish the presence of 10 transmembrane (TM) helices, suggest that the TMC proteins are dimers and reveal that the conserved $Ca^{2+}$ binding site found in TMEM16 proteins is not conserved in TMC. Each TMC1 protomer contains a large cavity at the periphery of the protein that is formed by the TM4 to TM7 helices and contains the *Bth* mutation, raising the possibility that it functions as an ion permeation pathway. To test the prediction that the MET channel contains an unusually large permeation pathway, we investigated the permeability of hair cells to fluorescently-labeled dextrans and we provide evidence that dextrans as large as 3 kDa can permeate. Dextran permeation is abolished by breaking tip links, blocking the MET channel or genetic deletion of TMC1/TMC2 proteins, suggesting that functional MET channels are required. Taken together, our results provide a structural framework for investigating TMC proteins, suggest the presence of a large permeation pathway and support the hypothesis that TMC1 is a pore forming subunit of the MET channel complex.

## Results

### Structural relationship between TMC1 and TMEM16 proteins

We began by investigating whether the application of hidden Markov model (HMM)-based profiles would allow the identification of suitable templates to model the structure of TMC1. HMM-based profiles are considered a superior tool to detect distantly related sequences in databases, improving the identification of valid templates (*Remmert et al., 2011*). Using this approach, nhTMEM16 (*Brunner et al., 2014*)(PDB ID: 4WIS) and mTMEM16A (*Paulino et al., 2017*)(PDB IDs: 5OYB and 5OYG) stand out as the best templates for TMC1 when compared to other candidates (*Figure 1—figure supplement 1*). The initial sequence alignments of mTMC1 with nhTMEM16 and mTMEM16A cover 80% of the sequences, which is remarkable given the limited sequence identity of 12% and 11%, respectively (*Figure 1—figure supplements 1* and *2A,B*). Protein hydrophobicity profiles are

evolutionarily better preserved than protein sequences, and proteins displaying similar hydrophobicity profiles tend to present a common fold (*Lolkema and Slotboom, 1998*; *Sarkar and Kellogg, 2010*). Interestingly, TMC1 and TMEM16 hydrophobicity profiles showed similar peak widths and distributions (*Figure 1A,B*), suggesting that TMC1 and TMEM16 proteins are relatedand that TMEM16 proteins are valid templates to model the structure of TMC1 despite their low sequence identity.

An accurate sequence alignment of TMC1 with both templates, nhTMEM16 and mTMEM16A, is essential to generate valid homology models. Since structure-guided alignments result in more accurate models (*Forrest et al., 2006*), we first superimposed the structures of nhTMEM16 and mTMEM16A (Ca$^{2+}$-bound state) to obtain an initial sequence alignment (*Figure 1—figure supplement 2C,D*), which was then combined with a HMM profile alignment of mTMC1 and nhTMEM16 (*Figure 1—figure supplement 2A*) to obtain an alignment of all three proteins. Initial sequence alignments were further refined by removing gaps within secondary structural elements and using the conservation scores from Consurf (*Ashkenazy et al., 2016*) as a guide to position highly-conserved residues within the protein core. The refined alignments showed a good correspondence between secondary structural elements, not only between the 10 TM helices but also within four structured loops (*Figure 1C*). The broad sequence coverage of both TMEM16 templates, the good correspondence between hydrophobicity profiles, and the strong correlation between TM helices in TMC1 and TMEM16 proteins indicate that these proteins exhibit a common fold.

## Generation and evaluation of TMC1 models

The refined sequence alignment of TMC1, nhTMEM16 and mTMEM16A (*Figure 1C*), the crystallographic structure of nhTMEM16 (*Brunner et al., 2014*)(PDB ID: 4WIS) and the cryo-EM structures of mTMEM16A in the Ca$^{2+}$-bound and Ca$^{2+}$-free state (*Paulino et al., 2017*)(PDB IDs: 5OYB and 5OYG) were used to generate three TMC1 structural models (model 1, model 2, and model 3, respectively) (*Figure 2A* and *Figure 2—figure supplement 1A–C*). As expected, these three models displayed an overall similar architecture, consisting of 10 TM helices, two helices in the N-terminus, one helix in the C-terminus, three long extracellular loops (TM1-TM2, TM5-TM6 and TM9-TM10) and one long intracellular loop (TM2-TM3) (*Figure 2A*). The predicted transmembrane topology of the TMC1 protein is distinct from earlier models containing between 6 and 8 TMs (*Fettiplace, 2016*; *Holt et al., 2014*; *Kawashima et al., 2015*; *Labay et al., 2010*), providing a revised framework to identify extracellular glycosylation sites and intracellular regulatory motifs (*Figure 2—figure supplement 2*). Our TMC1 models were generated as dimers, similar to the templates used, with a two-fold symmetry axis located between the two TM10 helices (*Figure 2—figure supplement 1A–C*). Residues in TM10 forming the dimer interface are highly conserved, consistent with TMC1 forming dimers (*Figure 3—figure supplement 1*). Each protomer contained a cavity formed by helices TM4-TM7 that faces the membrane. The major divergences among the three models were dictated by the structural templates, which exhibited unique rearrangements of helices TM4 and TM6 that contour the cavity (*Figure 2B*). The cavity observed in model 1 was wide open, forming a deep cleft towards the lipid bilayer, whereas in model 2, the cavity was narrower at the extracellular side and partially shielded from the membrane. In model 3, the intracellular end of the TM6 helix bends over the cavity to further diminish the exposure of the cavity to the surrounding membrane.

The stereochemical quality of each model was analyzed using PROCHECK (*Laskowski et al., 1993*), which showed two or fewer residues in disallowed regions of the Ramachandran plot for all three models. These residues were in the intracellular TM8-TM9 loop, indicating that the overall protein fold has good stereochemistry. In addition, we calculated the ProQM score averaged by the number of residues (global ProQM) (*Ray et al., 2010*) to assess the compatibility between the mTMC1 sequence and its fold in the three models. The global ProQM score of model 1 was 0.612, higher than the scores of model 2 (0.577) and model 3 (0.564). As is typical, these values were slightly lower than those obtained for the corresponding template structures: 0.662 for nhTMEM16, 0.632 for mTMEM16A in the Ca$^{2+}$-bound and 0.634 for mTMEM16A in the Ca$^{2+}$-free state. The comparable global ProQM scores between the models and the corresponding templates and the notable stereochemistry revealed by the PROCHECK analysis highlights the good quality of the models, further supporting the structural relationship between TMC and TMEM16 proteins.

We also used the ProQM score calculated at each residue position (local ProQM) (*Ray et al., 2010*) to evaluate the quality of each model near the cavity, where the 3-dimensional arrangement

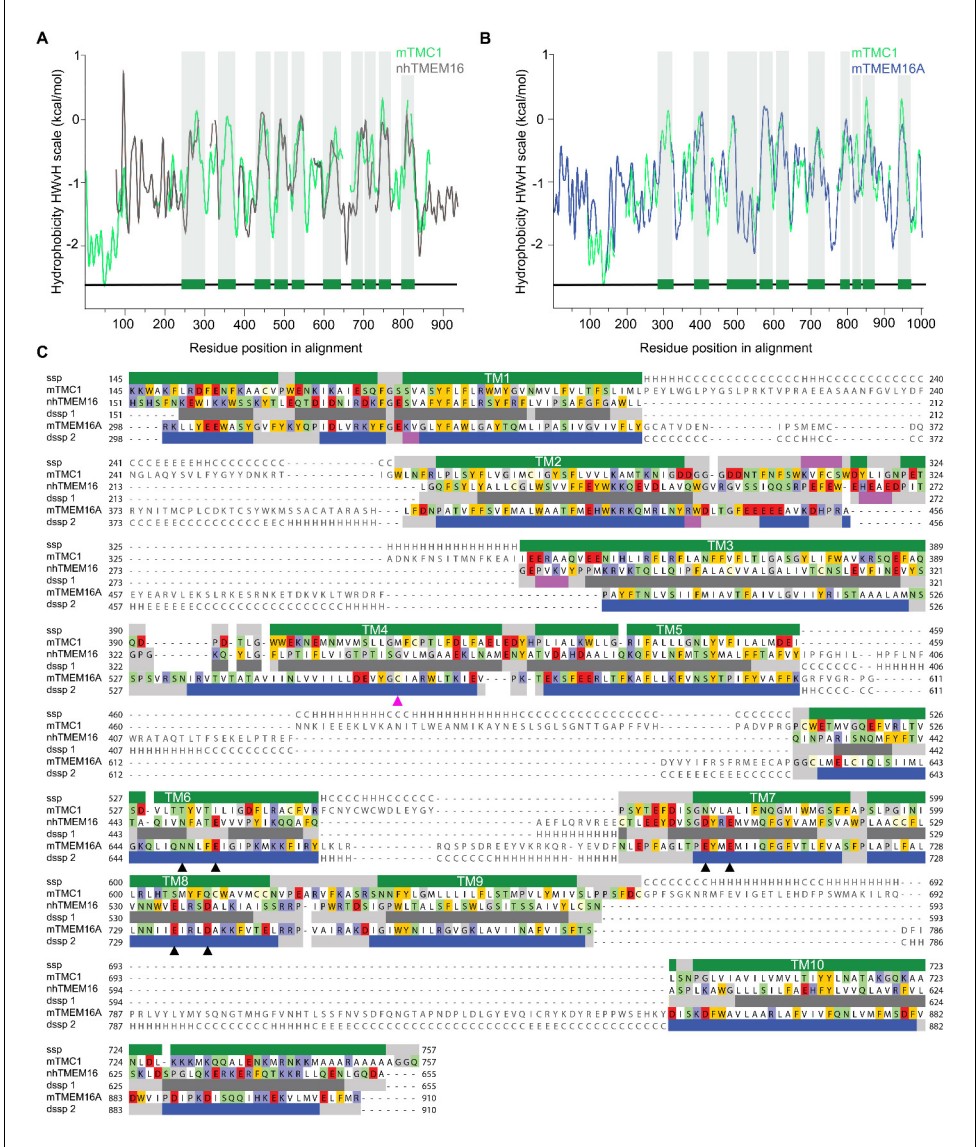

**Figure 1.** TMC1 and TMEM16 proteins share a common fold. (**A**) Hydrophobicity profiles for nhTMEM16 (gray) and mTMC1 (green) revealing conserved hydrophobic regions. Residues corresponding to mTMC1 TM helices are indicated with green bars and gray shading. (**B**) Hydrophobicity profiles for mTMEM16A (blue) and mTMC1 (green) revealing conserved hydrophobic regions. TM regions are indicated as in (**A**). (**C**) mTMC1, nhTMEM16 and mTMEM16A sequence alignment used to generate TMC1 models. Non-modeled residues are shown in gray without highlighting, while residues shown in the model are highlighted according to their properties; neutral in white, aromatic in yellow, polar in green, basic in blue and acidic in red. TMC1 secondary structure prediction (ssp) is shown at the top of the alignment and secondary structure information extracted from nhTMEM16 and mTMEM116A structures (dssp1 and dssp2, respectively) is included below the corresponding sequences. Grey rectangles indicate loops, pink rectangles β-strands, and α−helices are shown as green rectangles for TMC1, grey for nhTMEM16 and blue for mTMEM16A. Conserved TM helices are labeled and the six residues involved in calcium binding found in TMEM16 and absent in TMC proteins, are indicated with black arrowheads. The magenta arrowhead indicates the position of the Beethoven mutation in TMC1.

DOI: https://doi.org/10.7554/eLife.38433.002

The following figure supplements are available for figure 1:

**Figure supplement 1.** Results from HHpred search for TMC1 templates.

DOI: https://doi.org/10.7554/eLife.38433.003

**Figure supplement 2.** Initial sequence alignments.

DOI: https://doi.org/10.7554/eLife.38433.004

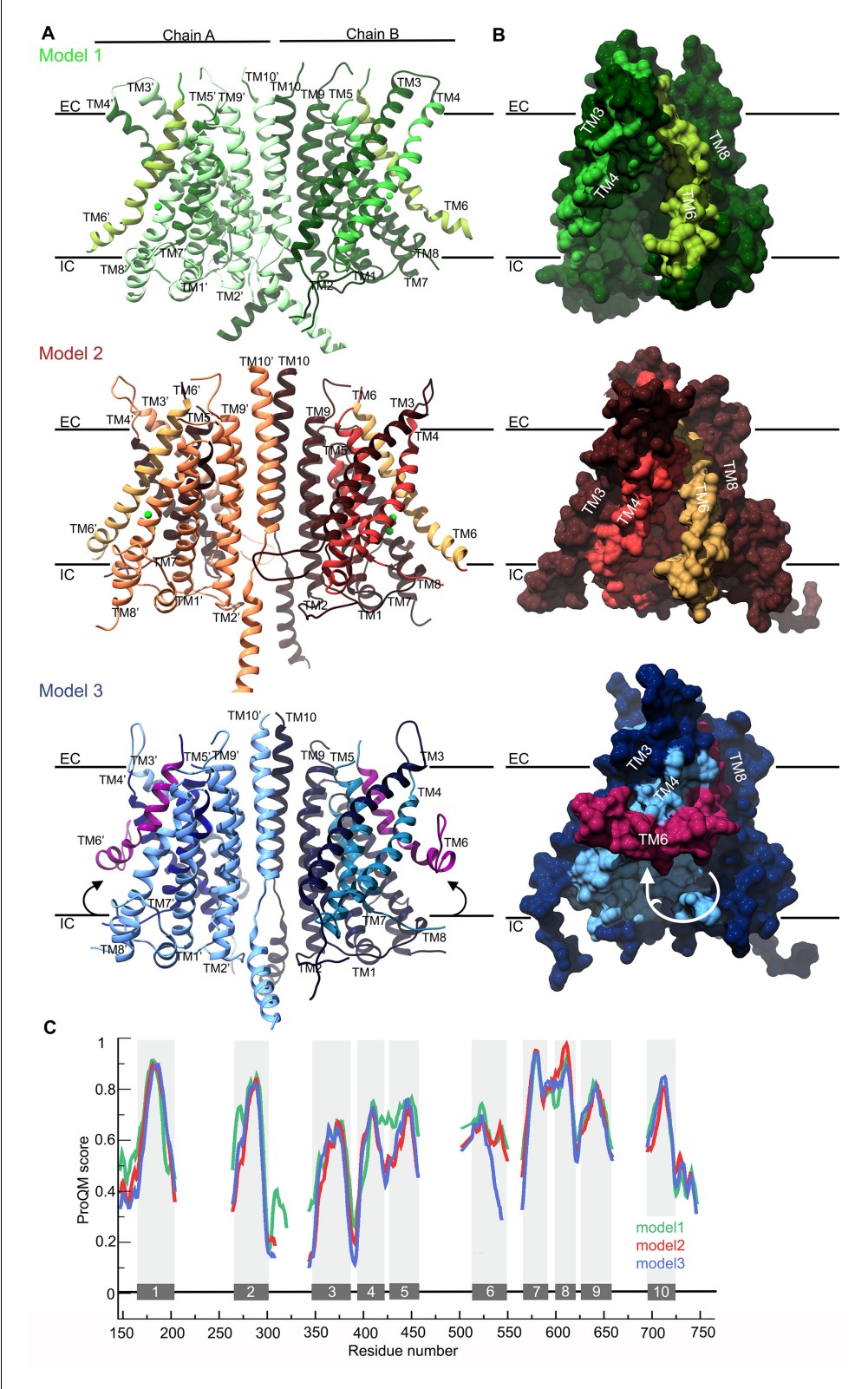

**Figure 2.** TMC1 models based on TMEM16 structures. (**A**) Ribbon representation of TMC1 models based on nhTMEM16 (model 1, green), mTMEM16A in the Ca²⁺-bound state (model 2, red) and mTMEM16A in the Ca²⁺-free state (model 3, blue). Chains A and B are indicated and colored in lighter and darker color, respectively. TM4-TM7 helices that form the cavity are colored in a different color and the TM6 is highlighted. The approximate position of the plasma membrane is indicated by black lines and the extracellular (EC) and intracellular (IC) sides are labeled. (**B**) 90° view of a TMC1 protomer from

*Figure 2 continued on next page*

*Figure 2 continued*

each model in A shown in surface representation, revealing the cavity formed by helices TM4-TM7. Model 1 reveals a wide-open cavity exposed to the membrane environment, while the TM4 helix is repositioned in model 2 to partially close the cavity (red) towards the EC side. In model 3, the TM6 helix (magenta) is bent, which further closes the cavity. (C) Local ProQM score revealing the main differences between the models, which are caused by the rearrangement of TM4 in model 2 (red) and TM6 in model 3 (blue). The 10 TM helices are indicated and represented with grey bars and shading.

DOI: https://doi.org/10.7554/eLife.38433.005

The following source data and figure supplements are available for figure 2:

**Source data 1.** TMC1 models 1, 2 and 3.
DOI: https://doi.org/10.7554/eLife.38433.008
**Figure supplement 1.** Structural superimposition of TMC1 models and TMEM16 templates.
DOI: https://doi.org/10.7554/eLife.38433.006
**Figure supplement 2.** Topology of TMC1.
DOI: https://doi.org/10.7554/eLife.38433.007

of the helices differs the most between models. We found that the local ProQM scores of residues located in TM4-6 were higher in model 1 compared to model 2 and model 3 (*Figure 2C*), indicating that TMC1 is more likely to contain a deep cavity, similar to that observed in the nhTMEM16 structure. Importantly, these local ProQM differences in helices TM4-TM6 were not detected in the template structures (*Figure 2—figure supplement 1D*), suggesting that the higher score of model 1 was not due to a trend in the ProQM at the template level, but to a higher compatibility of the mTMC1 sequence with the nhTMEM16 structure.

The cavity we observe in the TMC1 model 1 is of particular interest because it might contribute to the ion permeation pathway of the MET channel, similar to its role in the TMEM16 $Ca^{2+}$-activated $Cl^-$ channel and lipid scramblase (*Brunner et al., 2014*, *Brunner et al., 2016*; *Hartzell and Whitlock, 2016*; *Malvezzi et al., 2013*; *Paulino et al., 2017*; *Terashima et al., 2013*; *Whitlock and Hartzell, 2016*; *Yu et al., 2015*)(*Figure 3*). Four helices in TMC1 model 1 contribute to forming the cavity that faces the membrane environment, with the more hydrophobic TM5 and TM7 helices located deep within the cavity, and TM4 and TM6 helices located more peripherally. TM4 and TM6 contain many polar residues, including seven acidic residues, two basic residues and nine uncharged polar residues, giving the cavity considerable hydrophilic character (*Figure 3C*). Interestingly, TMC1 lacks the six acidic residues located behind the cavity in TMEM16 proteins that coordinate $Ca^{2+}$ and are required for activation (*Brunner et al., 2014*; *Tien et al., 2014*)(*Figure 1C*; *Figure 2—figure supplement 1A,B*). The surface of the cavity in TMC1 model 1 exhibits a negative electrostatic potential (*Figure 3D*), consistent with the cation selectivity observed for the MET channel (*Corey and Hudspeth, 1979*). Importantly, residues lining the cavity in TMC1 model 1 show strong sequence conservation among TMC proteins (*Figure 3E* and *Figure 3—figure supplement 1*), suggesting that the cavity is a conserved feature of these proteins. Indeed, the TMC domain (residues 512–631 in mTMC1, NP_083229.1) is the most highly conserved region among TMC homologs (*Kawashima et al., 2015*; *Kurima et al., 2003*) and this domain is formed by the TM6-TM8 helices, all of which contribute to forming the cavity.

Out of the thirty missense deafness-causing TMC1 mutations identified to date, twenty human and two mouse mutations are within regions we modeled (*Figure 4* and *Figure 4—source data 1* and *Figure 4—figure supplement 1*). Interestingly, nine of these mutations are found in TM4 and TM5, two of the helices that form the cavity. In particular, two positions where mutations cause autosomal dominant deafness are found in the cavity, including the M412K *Bth* mutant (M418K in hTMC1) and D569N/H (D572N/H in hTMC1) (*Figure 4*, *Figure 4—source data 1* and *Figure 4—figure supplement 1*). Taken together, the localization of these deafness-causing mutations to the cavity in TMC1 raises the possibility that it functions as an ion permeation pathway.

We wondered whether we could take advantage of the structural relationship between TMC and TMEM16 proteins to generate chimeric constructs that traffic to the plasma membrane when expressed in mammalian cells, as has been demonstrated for TMEM16A (*Duran et al., 2012*; *Yu et al., 2015*), yet retain the cavity of TMC1 and thereby might be functional ion channels. We generated two chimeras between mTMC1 and mTMEM16A, one where we transplanted the N- and C-terminal regions of TMEM16A into TMC1 (chimera 1), and a second where we transplanted the core of the TMC1 protein containing the cavity (TM3-8 helices) into TMEM16A (chimera 2). We could

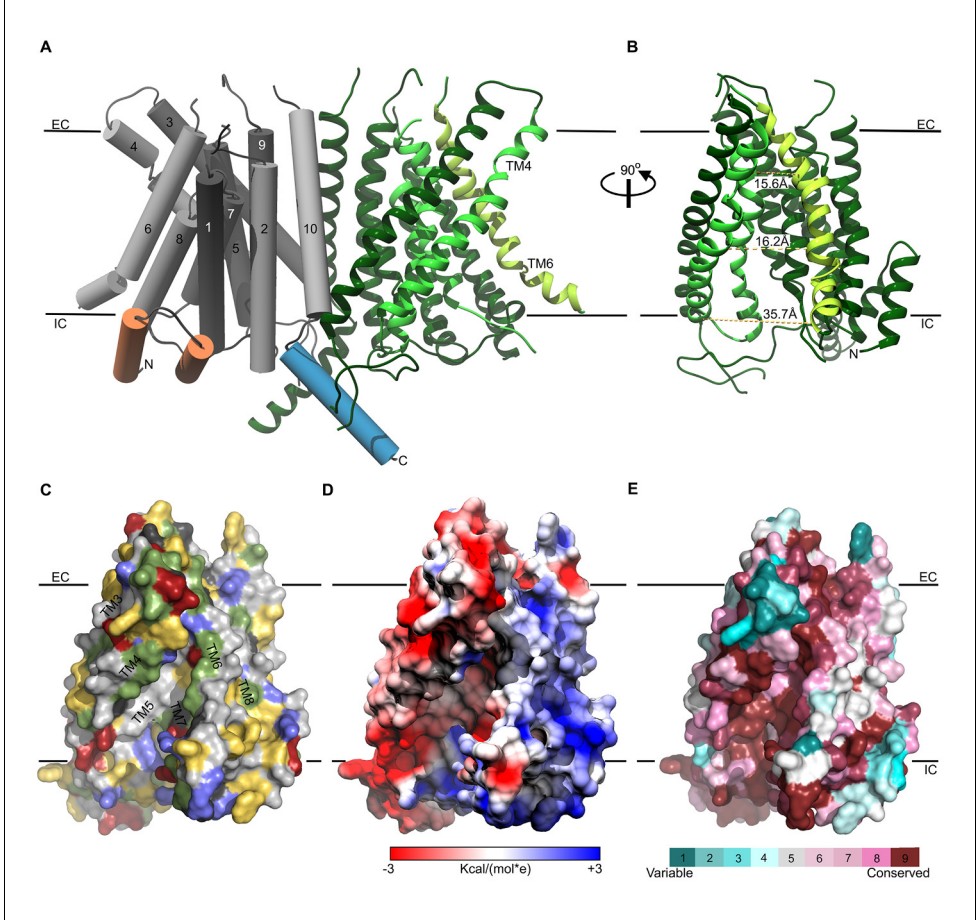

**Figure 3.** Representation of the TMC1 model 1. (**A**) The TMC1 dimer is represented with the α-helices of one protomer as cylinders with TM helices numbered and colored in grey, the two intracellular N-terminal α−helices are colored in orange and the C-terminal α−helix is colored in blue. The other TMC1 protomer is shown in ribbons representation with the TM4-TM7 that form the cavity colored in lighter green and TM6 in lime. TM4 and TM6 lining the cavity are indicated. Plasma membrane position is indicated by black lines and the extracellular (EC) and intracellular (IC) sides are labeled. (**B**) 90° view of one TMC1 protomer revealing the cavity formed by the TM4-TM7 helices. Indicated distances are between alpha-carbons of residues located on TM4 and TM6: R523-W397 (15.6 Å), S404-T535 (16.2 Å), and V547-E423 (35.7 Å). (**C**) Surface representation of the cavity in one TMC1 protomer (as in **B**), color-coded based on residue properties: neutral-white, aromatic-yellow, polar-green, basic-blue, and acidic-red. (**D**) TMC1 cavity colored based on its electrostatic potential. (**E**) TMC1 cavity with residues colored based on their conservation score.
DOI: https://doi.org/10.7554/eLife.38433.009

The following figure supplement is available for figure 3:

**Figure supplement 1.** TMC1 residue conservation.
DOI: https://doi.org/10.7554/eLife.38433.010

readily observe GFP-tagged TMEM16A express and traffic to the plasma membrane of HEK293 cells, but for myc-tagged constructs of TMC1 and the two chimeras, we observed only intracellular localization (*Figure 4—figure supplement 2*).

## Hair cells uptake texas red-labeled 3 kDa dextran

The structural models we generated for TMC1 raise the possibility that this protein contains a permeation pathway that is localized near the protein-lipid interface, as has been proposed for both the TMEM16 $Ca^{2+}$-activated $Cl^-$ channel and lipid scramblase (*Jiang et al., 2017*; *Lee et al., 2016*; *Whitlock and Hartzell, 2016*; *Yu et al., 2015*). It is known that inner ear hair cells are permeable to relatively large cationic molecules. For instance, amphipathic styryl dyes (FM1-43 and FM4-64, 0.5–0.8 kDa) and Texas Red-labeled aminoglycosides (1.29–1.43 kDa) label the hair cell body, and both molecules are thought to enter through the MET channel (*Alharazneh et al., 2011*;

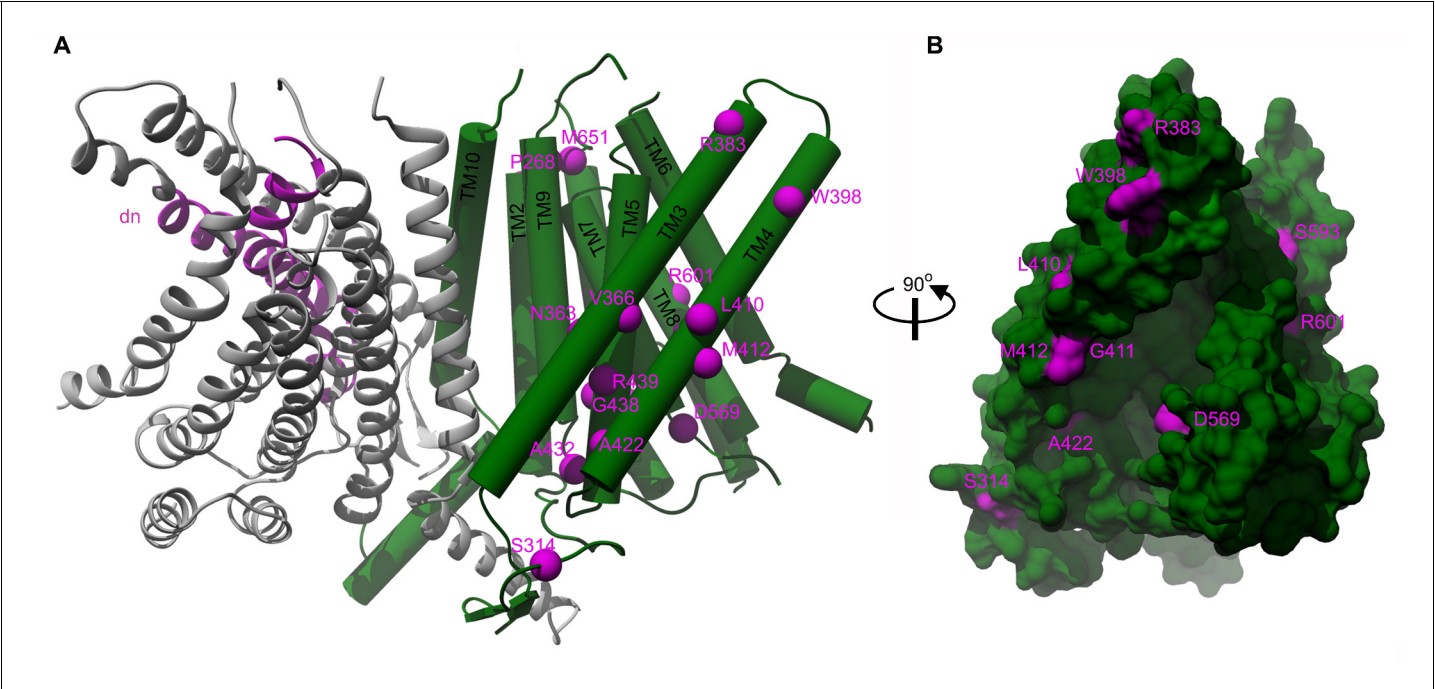

**Figure 4.** Localization of deafness-causing mutations in TMC1 model 1. (**A**) TMC1 model 1 with one protomer in ribbon representation and colored in grey showing the region deleted in the *Deafness (dn)* mouse model (Δ402–459) in magenta. The helices of the other protomer are represented as cylinders and the deafness-causing mutations are labeled and indicated as magenta spheres. (**B**) 90° degrees view of panel A in surface representation highlighting the M412, A422 and D569 mutations in the cavity.

DOI: https://doi.org/10.7554/eLife.38433.011

The following source data and figure supplements are available for figure 4:

**Source data 1.** TMC1 deafness-causing mutations.

DOI: https://doi.org/10.7554/eLife.38433.014

**Figure supplement 1.** Localization of deafness-causing mutations in TMC1 models.

DOI: https://doi.org/10.7554/eLife.38433.012

**Figure supplement 2.** TMC1 and TMC1-TMEM16A chimeras do not traffic to the plasma membrane when expressed in HEK293T cells.

DOI: https://doi.org/10.7554/eLife.38433.013

*Kawashima et al., 2011*; *Marcotti et al., 2005*; *Meyers et al., 2003*). To explore the limits of the MET pore size and provide experimental support for the large cavity observed in our TMC1 model, we investigated whether cochlear hair cells might be permeable to Texas Red-labeled 3 kDa dextrans (dextran-TR) (*Figure 5A*), polar polymers of glucose molecules that are derivatized with Lys for fixation and making them cationic. When organ of Corti explants from wild-type postnatal day (P) 6 mice were incubated for 2 hr with dextran-TR at room temperature, we observed robust and specific labeling of auditory hair cells using confocal Airyscan super-resolution microscopy (*Figure 5*). Labeling was observed for both the inner and outer hair cells (IHC and OHC) along the entire organ of Corti from the basal to the apical regions (*Figure 5B–D*). Interestingly, we found that hair cells from the basal area incorporated significantly more dextran compared to those from the middle or the apical regions (*Figure 5C,D*), a result that is consistent with reports for uptake of FM1-43 (*Gale et al., 2001*; *Lelli et al., 2009*). Because we could not detect any differences in the dextran-TR uptake between OHC and IHC within any of the different cochlear regions examined, in subsequent experiments OHC and IHC were analyzed together. We also tested a shorter incubation times, and although we observed fluorescence accumulation in the hair cell body, the signals were weaker and more variable (*Figure 5—figure supplement 1*). It was interesting that not all hair cells in a given region efficiently accumulated dextran-TR (*Figure 5C–D*), a pattern that is also seen in previous studies with uptake of Texas Red-labeled aminoglycosides (*Hailey et al., 2017*; *Kawashima et al., 2011*; *Park et al., 2013*) and in studies examining the activity of hair cells with $Ca^{2+}$ indicators (*Spinelli and Gillespie, 2012*; *Zhang et al., 2018*). One of these studies also

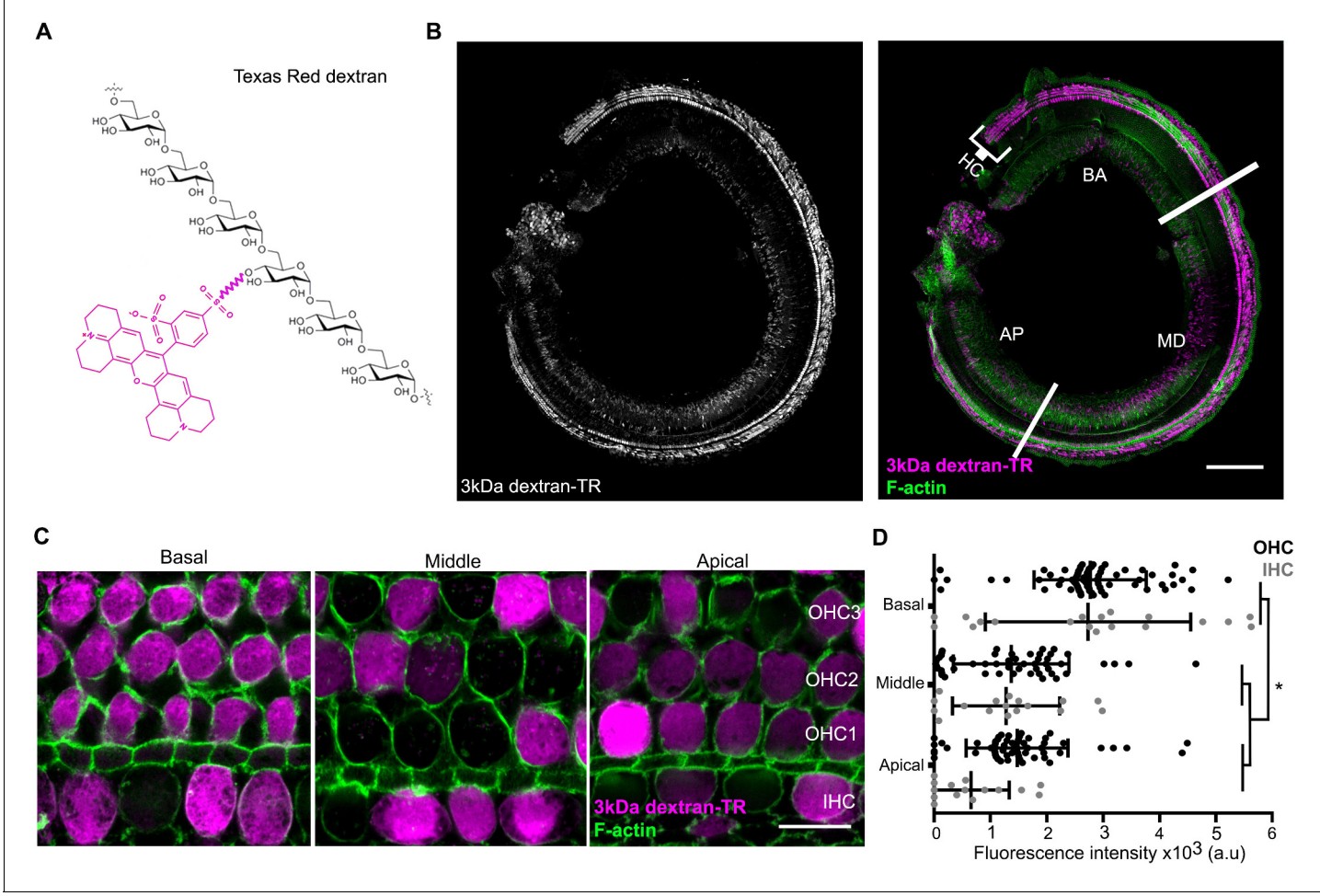

**Figure 5.** Hair cells uptake 3 kDa dextran-TR. (**A**) Schematic representation of Texas Redlabeled dextran containing six molecules of glucose corresponding to a molecular weight of 1.08 kDa. A molecule of Texas Red is shown in magenta linked by a succinimidyl ester reaction to a glucose monomer. (**B**) Confocal image showing specific sensory hair cells (HC) labeling of 3 kDa dextran-TR across the whole organ of Corti from a 6-day-old mouse. The basal (BA), middle (MD), and apical (AP) regions of the organ are indicated. Scale bar represents 200 µm. (**C**) 3 kDa dextran-TR fluorescence (magenta) intensity at the hair cell body from the basal, middle and apical regions of the organ of Corti. One representative image for each area is shown. The three rows of outer hair cells (OHC) and one row of inner hair cells (IHC) are indicated. Scale bar represents 20 µm. Phalloidin was used to label F-actin (green) and visualize stereocilia and hair cell boundaries in panel B and C. (**D**) Quantification of fluorescence intensity in the cell body of hair cells at the different cochlear regions (basal, n = 89; middle, n = 69 and apical, n = 74). Each dot represents a single hair cell (OHC in black; IHC in gray). Mean and standard deviation for each group are shown. Asterisks indicate significant differences among the different cochlear regions (p < 0.01, one-way ANOVA with Turkey´s multiple comparison test).

DOI: https://doi.org/10.7554/eLife.38433.015

The following source data and figure supplements are available for figure 5:

**Source data 1.** Mean fluorescence values for *Figure 5D*.
DOI: https://doi.org/10.7554/eLife.38433.018

**Figure supplement 1.** Time course of 3 kDa dextran-TR uptake into hair cells.
DOI: https://doi.org/10.7554/eLife.38433.016

**Figure supplement 1—source data 1.** Mean fluorescence values for *Figure 5—figure supplement 1B*.
DOI: https://doi.org/10.7554/eLife.38433.017

investigated the integrity of tip links using electron microscopy and attributed the variability to breakage of tip links during explant preparation (*Spinelli and Gillespie, 2012*).

## Labeling of stereocilia in hair cells that uptake dextran

If the uptake of dextran-TR results from permeation of the dye through the MET channel, we would expect it to then freely diffuse through the hair cell stereocilia into the cell body. When we imaged both stereocilia and cell bodies, we observed that only those cells that incorporated dextran-TR in their cell body also showed fluorescent labeling of their stereocilia (*Figure 6A*). Furthermore, hair cells whose stereocilia displayed the strongest signals also had the strongest fluorescence signals in their cell bodies (*Figure 6A*, bottom panels). Importantly, we observed a uniform diffuse fluorescence signal along the stereocilia with enrichment at the tips of the shorter stereocilia rows, a pattern that was more evident in IHC where the stereocilia are larger (*Figure 6B*). We also carried out a 3D reconstruction from a z-series of images and generated a volume representation of the dextran signal in an IHC (*Figure 6C*). Dextran-TR signal was found along the shorter stereocilia rows, cuticular plate, and an extended apical region. The tip of the tallest stereocilia row was also labeled in some cells but presented a weaker and dotted-like discontinuous intensity (*Figure 6C*). We also observed labeling of vesicle-like structures in neighboring supporting cells, suggesting that dextran-TR can be taken up by non-sensory supporting cells, possibly through endocytosis (*Figure 6C*). Altogether, these results indicate that there is an intimate relationship between dextran-TR labeling of the stereocilia and of the cell body of hair cells.

## Dextran uptake at lower temperature

In addition to the diffuse pattern of labeling of hair cell bodies described thus far, we also see labeling of vesicle-like structures within the cell bodies of hair cells that have a punctate appearance, similar to what is seen in supporting cells (see OHC in *Figure 6B*). Interestingly, this punctate labeling of hair cell bodies is more readily discernable in hair cells that do not display diffuse labeling (compare the two OHC bodies shown in *Figure 6B*), suggesting an endocytic mechanism of uptake. To test whether the diffuse labeling pattern observed in cell bodies is due to endocytosis, we compared dextran uptake at 22 and 4°C, a temperature at which endocytosis is repressed. We observed robust diffuse labeling of both cell bodies and stereocilia with dextran-TR at 4°C (*Figure 7A–C*), suggesting that the diffuse labeling of hair cells by dextran-TR does not require endocytosis.

## Dextran uptake is dependent on fluorophore charge and dextran size

To investigate the extent to which dextran uptake into hair cells is dependent on the size and net charge of the fluorophore, we tested for uptake of the larger 10 kDa dextran-TR, as well as both 3 and 10 kDa dextrans labeled with negatively charged fluorescein (dextran-FITC). In contrast to the diffuse pattern of uptake observed with 3 kDa dextran-TR, 10 kDa dextran-TR showed only punctate vesicle-like structures in the hair cell body. Moreover, the 10 kDa dextran accumulated around the hair cell membrane in a patchy pattern regardless of whether the label was Texas Red or anionic FITC (*Figure 7—figure supplement 1A*). In addition, vesicle-like structures were seen in supporting cells (*Figure 7—figure supplement 1A*). Interestingly, the same vesicle-like pattern was observed when cells were incubated with anionic 3 kDa dextran-FITC (*Figure 7—figure supplement 1B*), suggesting that the properties of the fluorophore influence whether a diffuse pattern of uptake is observed. FITC is actually smaller than Texas Red, so most likely the absence of diffuse uptake observed with FITC is related to the negative charge on the fluorophore or its increased hydrophilicity. The size-dependence for dextran uptake demonstrates specificity to the diffuse pattern of uptake observed with 3 kDa dextran-TR, and shows that a vesicle-like pattern of uptake can be distinguished from the more diffuse pattern.

To test whether the diffuse pattern of uptake observed with 3 kDa dextran-TR is caused by cell damage during the preparation of the explant, we examined the uptake of 3 kDa dextran-TR and 10 kDa dextran-FITC simultaneously. As in the experiments described thus far, we observed both diffuse uptake and a vesicle-like pattern of hair cell body labeling for the 3 kDa dextran-TR. However, only a vesicle-like pattern of uptake was observed for 10 kDa dextran-FITC (*Figure 7—figure supplement 1C*). The vesicle-like structures observed with both 10 kDa dextran-FITC and with 3 kDa dextran-TR suggest that this form of uptake likely reflects endocytosis. From these results, we

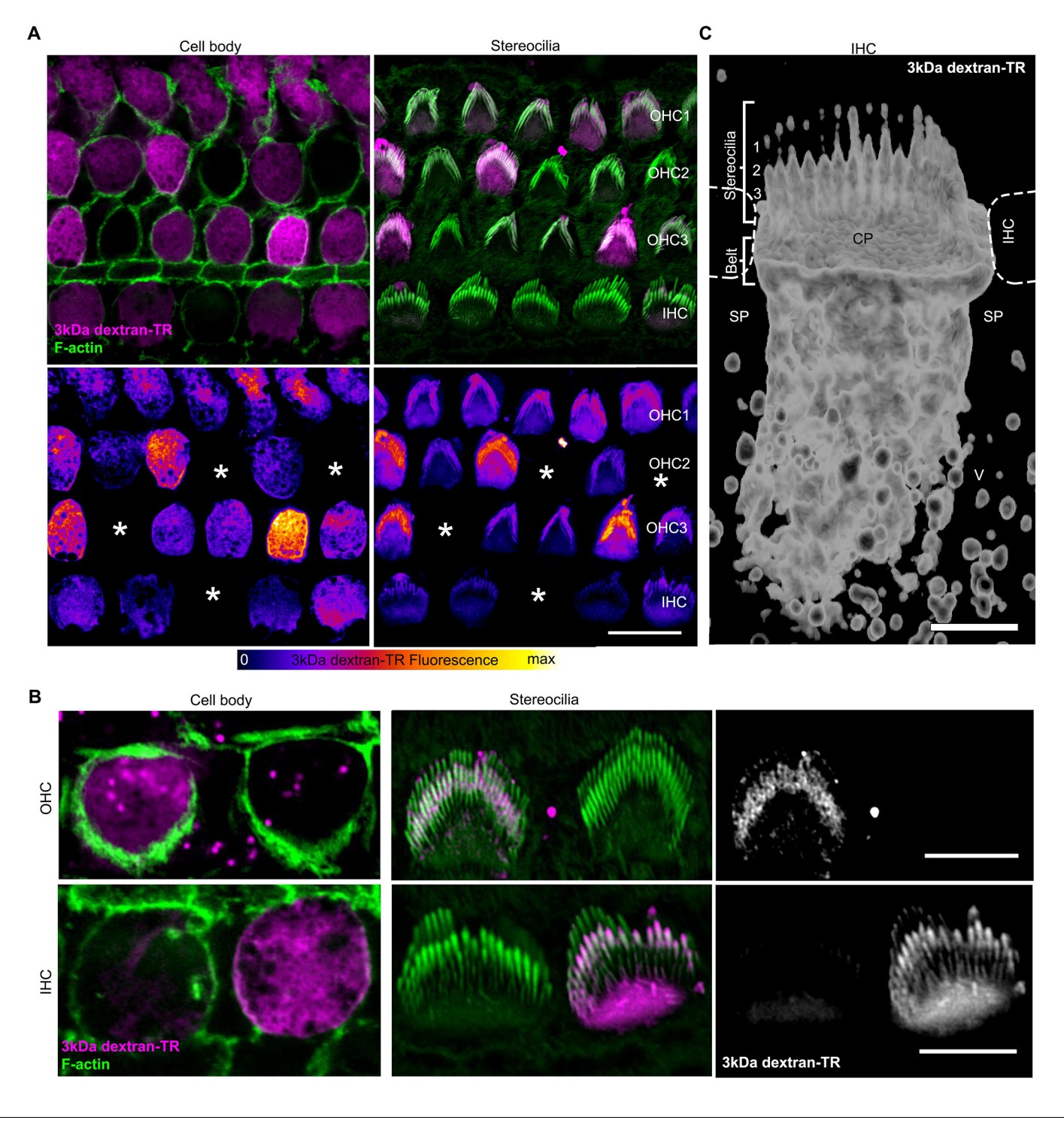

**Figure 6.** Stereocilia labeling in hair cells taking up 3 kDa dextran-TR. (**A**) Confocal images displaying 3 kDa dextran-TR fluorescence (magenta in top panels) at the cell body and stereocilia counterstained with phalloidin to label F-actin (green) and thereby visualize hair cell boundaries and stereocilia. 3 kDa dextran-TR fluorescence from the same confocal images is shown below represented with the fire-LUT scale, which displays the amount of 3 kDa dextran-TR fluorescence in different colors (low in blue, medium in red and high in yellow). Cells lacking 3 kDa dextran-TR fluorescence at the cell body and stereocilia are indicated with an asterisk. Scale bar represents 20 μm. (**B**) Closer view of 3 kDa dextran-TR fluorescence at the cell body and stereocilia of outer hair cells (OHC) and inner hair cells (IHC). Scale bar represents 5 μm. (**C**) 3-D volume reconstruction of the 3 kDa dextran-TR

*Figure 6 continued on next page*

*Figure 6 continued*
fluorescence detected in a z-series image of the apical region of an IHC . Hair cells rows are numbered. Neighboring IHC, cuticular plate (CP) and supporting cells (SP) vesicles (V) are also indicated. Scale bar represents 2 µm.
DOI: https://doi.org/10.7554/eLife.38433.019

concluded that small dextran-TR enters the hair cells through two pathways, one common to other dextrans and involving vesicle-like structures, and another that does not involve endocytosis and producing a diffuse pattern of uptake.

### Disruption of tip links and MET channel blockers prevent uptake of dextran-TR into hair cells

To explore whether the non-endocytic pathway for uptake of 3 kDa dextran-TR might be the pore of the MET channel, we assessed dextran incorporation in hair cells from explants incubated with MET channel blockers or with the $Ca^{2+}$ chelator BAPTA, a manipulation that abolishes the mechanotransduction current in hair cells by breaking the tip links (*Assad et al., 1991*; *Fettiplace and Kim, 2014*; *Marcotti et al., 2005*; *Rüsch et al., 1994*). BAPTA and the MET channel blockers, dihydrostreptomycin (DHS), benzamil, neomycin, and amiloride completely abolish the diffusive pattern of uptake for 3 kDa dextran-TR (*Figure 8A,B*). The stereocilia labeling observed in control samples was also

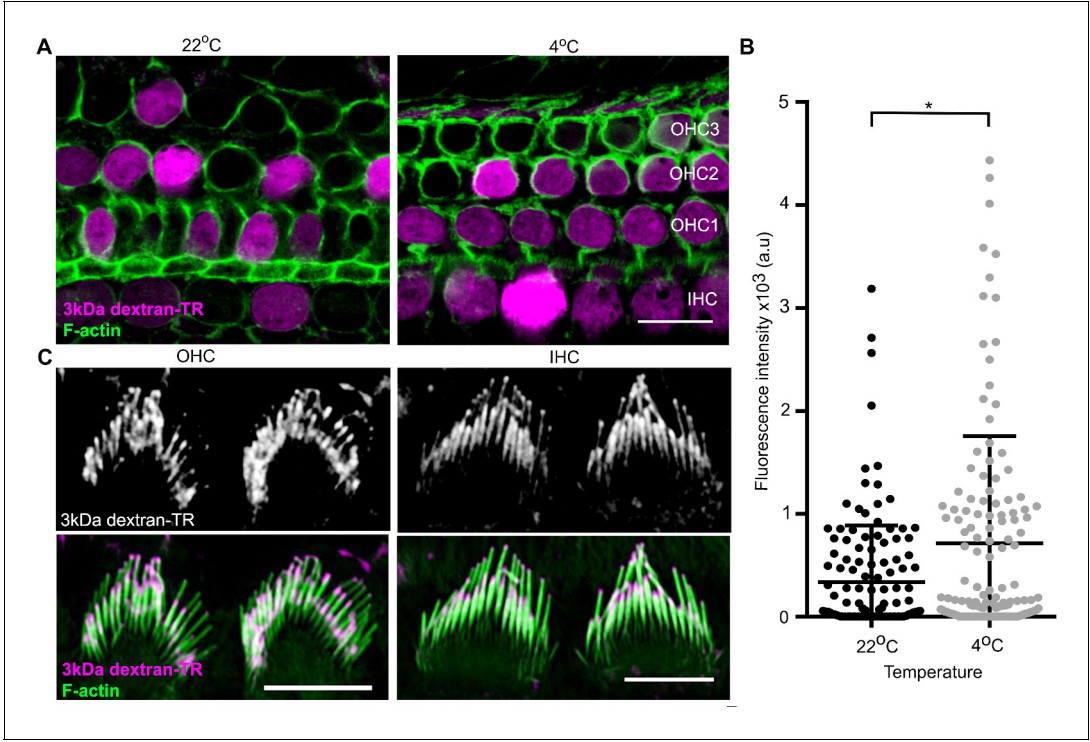

**Figure 7.** Dextran-TR uptake is maintained at low temperature. (**A**) Representative confocal images displaying 3 kDa dextran-TR accumulation (magenta) in mice hair cells at 4 and 22°C. The scale bar represents 20 µm. (**B**) Fluorescence intensity quantification in the cell body of hair cells at 22°C (black, n = 136) or 4°C (gray, n = 140). Each dot represents a single hair cell and the mean and standard deviation for each condition are shown. Asterisk indicates significant increase in fluorescence intensity at 4°C (p < 0.01, unpaired t-test). (**C**) 3 kDa dextran-TR localization at the stereocilia of OHC (left) and IHC (right) after 2 hr incubation at 4°C. The scale bar represents 5 µm.
DOI: https://doi.org/10.7554/eLife.38433.020

The following source data and figure supplement are available for figure 7:

**Source data 1.** Mean fluorescence values for *Figure 7B*.
DOI: https://doi.org/10.7554/eLife.38433.022
**Figure supplement 1.** Testing for uptake of 10 kDa dextrans and 3 kDa dextran-FITC.
DOI: https://doi.org/10.7554/eLife.38433.021

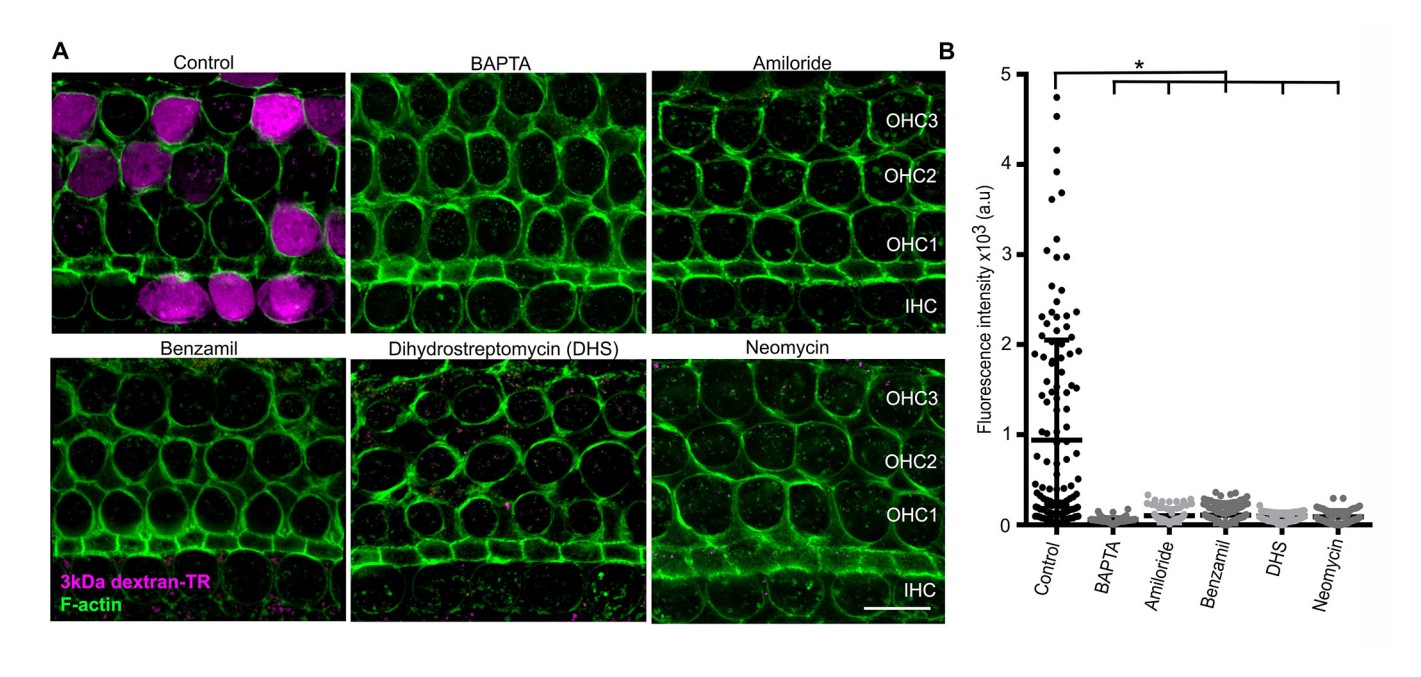

**Figure 8.** Breaking tip-links and blockade of the MET channel abolish dextran uptake. (**A**) Representative confocal images of hair cells after incubation with 3 kDa dextran-TR in the absence (control) or presence of BAPTA (5 mM) or MET channel blockers; amiloride (150 µM), benzamil (30 µM), neomycin (500 µM), dihydrostreptomycin (500 µM). 3 kDa dextran-TR fluorescence is shown in magenta and tissue was counterstained with phalloidin for visualization of the hair cell boundaries (green). The three rows of outer hair cells (OHC) and one row of inner hair cells (IHC) are indicated and the scale bar represents 20 µm. (**B**) Quantification of 3 kDa dextran-TR fluorescence intensity in the cell body of hair cells in the absence (control, n = 133) or presence of BAPTA (n = 116), amiloride (n = 72), benzamil (n = 111), DHS (n = 64) or neomycin (n = 80). Each dot represents a single hair cell, and mean and standard deviation for each condition are shown. Asterisks indicate significant differences between the control and different blocker conditions (p< 0.01, one-way ANOVA with Turkey´s multiple comparison test).

DOI: https://doi.org/10.7554/eLife.38433.023

The following source data and figure supplement are available for figure 8:

**Source data 1.** Mean fluorescence values for *Figure 8B*.
DOI: https://doi.org/10.7554/eLife.38433.025

**Figure supplement 1.** Breaking tip-links and blockade of the MET channel both abolish dextran uptake and stereocilia labeling.
DOI: https://doi.org/10.7554/eLife.38433.024

not detectable in BAPTA or MET channel blockers-treated explants (*Figure 8—figure supplement 1A*), further supporting the link between stereocilia labeling and uptake of 3 kDa dextran-TR. In contrast, the vesicle-like pattern of labeling with 3 kDa dextran-TR was preserved in cells treated with BAPTA or MET blockers, with the notable exception of amiloride (*Figure 8—figure supplement 1B*), indicating that the vesicle-like pattern of uptake is independent of operational MET channels. Interestingly, amiloride is known to inhibit the Na$^+$-H$^+$ exchanger and thereby inhibit endocytosis (*Dutta and Donaldson, 2012*; *Koivusalo et al., 2010*; *Maréchal et al., 2001*; *West et al., 1989*), which would explain the absence of the vesicle-like pattern in hair cells treated with this blocker. Taken together, these data indicate that the stereocilia labeling and the diffuse pattern of dextran-TR uptake into hair cells requires functional MET channels.

## TMC1 and TMC2 proteins are essential for dextran-TR uptake

It has been reported that hair cells from TMC1/TMC2 double knockout (dKO) mice lack functional MET channels and do not uptake Texas Red-labeled aminoglycosides or FM1-43 dye (*Kawashima et al., 2011*, *Kawashima et al., 2015*). To examine the role of TMC proteins in dextran-TR uptake, we compared the uptake of 3 kDa dextran-TR in wild-type, single TMC1 knockout (KO), TMC2 KO, and TMC1/TMC2 dKO hair cells (*Figure 9* and *Figure 9—figure supplement 1A*). We observed a robust diffuse pattern of 3 kDa dextran-TR uptake into hair cells from wild-type

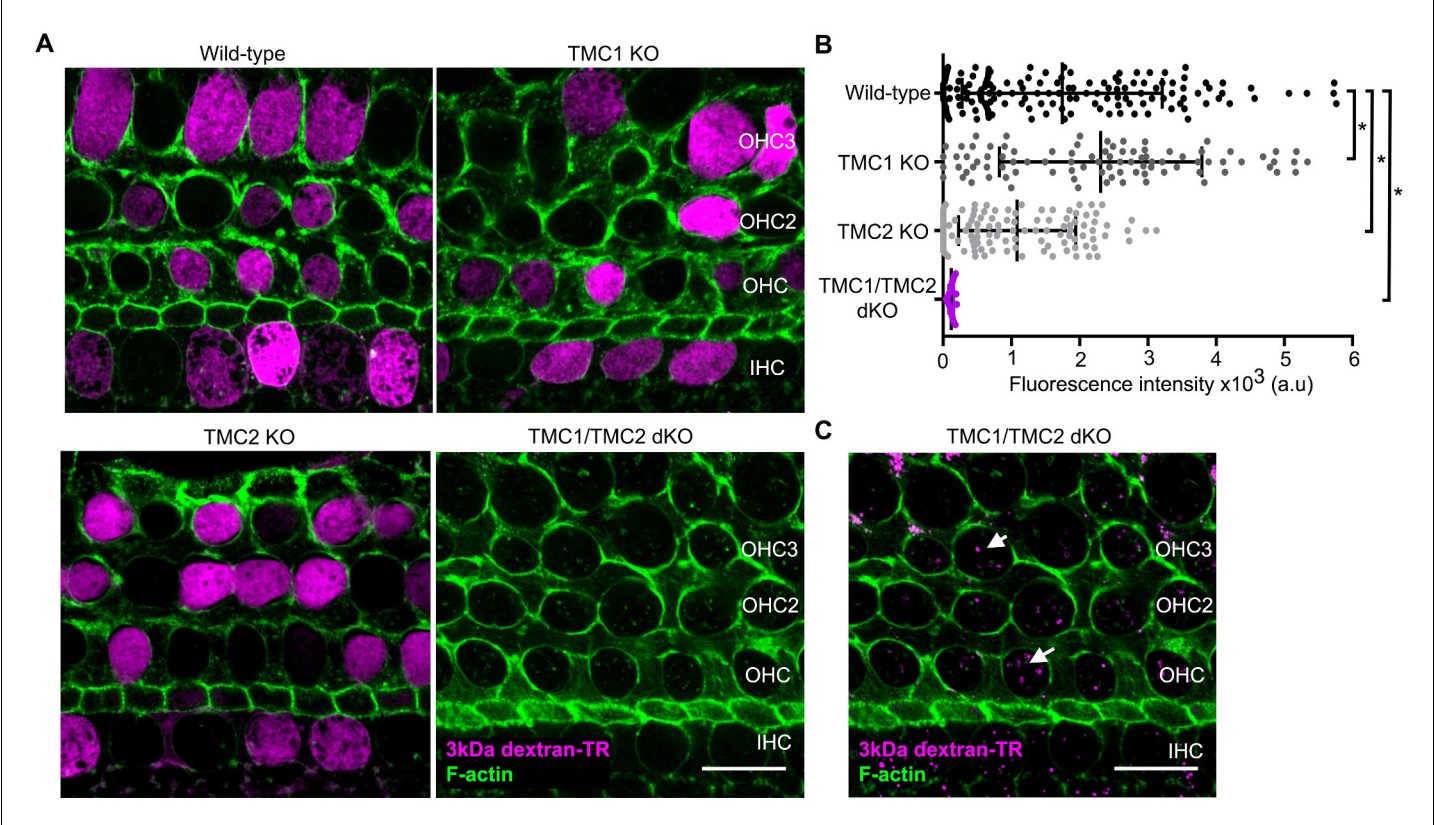

**Figure 9.** TMC1 and TMC2 are essential for the uptake of 3 kDa dextran-TR into hair cells. (**A**) Confocal images showing dextran-TR (magenta) uptake into hair cell bodies from wild-type, TMC1 KO, TMC2 KO or TMC1/TMC2 dKO mice. Phalloidin was used to counterstain for F-actin and visualize hair cell boundaries (green). Three rows of outer hair cells (OHC) and one row of inner hair cells (IHC) are indicated. Scale bar indicates 20 μm. (**B**) Quantification of fluorescence intensity in the cell body of wild-type (n = 146), TMC1 KO (n = 91), TMC2 KO (n = 111) and TMC1/TMC2 dKO (n = 94) mice. Each dot represents a single hair cell, and the mean and standard deviation for each group are shown. Asterisks indicate significant differences among the different mice genotypes (p < 0.01, one-way ANOVA with Turkey´s multiple comparison test). (**C**) A vesicle-like pattern is also observed with 3 kDa dextran-TR in TMC1/TMC2 dKO mice hair cells when the image display range was linearly adjusted in ImageJ.

DOI: https://doi.org/10.7554/eLife.38433.026

The following source data and figure supplement are available for figure 9:

**Source data 1.** Mean fluorescence values for *Figure 9B*.
DOI: https://doi.org/10.7554/eLife.38433.028

**Figure supplement 1.** TMC1 and TMC2 are essential for dextran-TR uptake and stereocilia labeling.
DOI: https://doi.org/10.7554/eLife.38433.027

mice, TMC1 KO, and TMC2 KO mice, consistent with the involvement of MET channels in dextran uptake because the hair cells from these mice display functional mechanotransduction currents at P6 (*Kawashima et al., 2011*, *Kawashima et al., 2015*). In contrast, we could not detect the diffuse pattern of 3 kDa dextran-TR uptake into hair cells from TMC1/TMC2 dKO mice (*Figure 9A,B*). Importantly, labeling of the stereocilia was also not detectable in TMC1/TMC2 dKO hair cells, whereas a strong dextran-TR signal was observed in the stereocilia of hair cells from wild-type or single KOs (*Figure 9—figure supplement 1B*). These data further demonstrate a critical role of the MET channel in dextran-TR labeling of stereocilia and diffuse uptake into hair cells. Interestingly, when we increased the brightness of these images, we observed a vesicle-like pattern in hair cells from TMC1/TMC2 dKO mice, suggesting that the endocytic entry of dextran is preserved in the absence of the TMC proteins (*Figure 9C*). These results confirm the requirement of functional MET channel and suggest an essential role of TMC1 and TMC2 proteins for labeling of stereocilia by dextran-TR and the diffuse pattern of dextran-TR uptake into cochlear hair cell bodies.

## Discussion

The initial goal of this study was to evaluate the proposed structural relationship between TMC proteins and the TMEM16 family of proteins. TMC1 and TMC2 are strong candidates for forming the pore of the MET channel in hair cells, while TMEM16A and B are $Ca^{2+}$-activated $Cl^-$ channels and the TMEM16F is a phospholipid scramblase. Our HMM-based sequence alignments and hydrophobicity analysis (*Figure 1*), together with the homology models we built and assessed (*Figure 2*), establish that TMC and TMEM16 proteins adopt a similar fold. Our models suggest that the TMC proteins have a transmembrane cavity at the protein-lipid interface (*Figures 2* and *3*), a feature that is dictated by the structures we used as templates (*Figure 2—figure supplement 1*). In the case of the nhTMEM16 scramblase (model 1), this cavity is completely open to the lipid bilayer and is thought to be the pathway for lipids to diffuse across the membrane (*Brunner et al., 2014*; *Malvezzi et al., 2013*). In the case of the TMEM16A $Ca^{2+}$-activated $Cl^-$ channels, the cavity is partially closed to the surrounding membrane, primarily within the outer half of the bilayer (*Paulino et al., 2017*). Although our models are inherently low resolution, it is interesting that the model based on the nhTMEM16 structure exhibited the highest ProQM score, in particular for the TM4-6 helices (*Figure 2*), regions where the structures differ the most between nhTMEM16 and TMEM16A in the absence and presence of $Ca^{2+}$. The cavity in our model of the TMC1 protein is wide and anionic (*Figure 3*), consistent with the possibility that this region contributes to forming the permeation pathway for the cation selective MET channel. It is also interesting that the cavity in our TMC1 models is highly conserved across TMC proteins (*Figure 3*), supporting a critical role in the structure and function of the protein. In addition, many mutations in TMC1 that cause deafness localize near the peripheral cavity, with several lining the putative permeation pathway (*Figure 4*, *Figure 4—figure supplement 1*). Notably, the only two positions where mutations lead to autosomal dominant progressive deafness in humans (DFNA36) are the *Bth* mutation (mTMC1 M412) and two mutations in residue D572 (mTMC1 D569), and both of these positions line the cavity in our TMC1 model (*Figure 4*).

Our model of the TMC1 protein provides a framework for further exploring the structural interactions between the components of the MET channel complex. For example, the peripheral location of the cavity in TMC1 raises the possibility that TM regions of TMHS, TMIE and/or PCDH15 might contribute to forming the permeation pathway together with TMC1. The homodimeric architecture of our TMC1 model is also interesting because the tip link is a homodimer of PCDH15 (*Kachar et al., 2000*; *Kazmierczak et al., 2007*) and each tip link is thought to bind to two MET channels (*Beurg et al., 2006*, *Beurg et al., 2009*; *Ricci et al., 2003*). PCDH15 contains a single TM helix and its intracellular C-terminus has been shown to interact with the N-terminus of TMC1 (*Maeda et al., 2014*), which in our model is positioned on the intracellular surface below the cavity that opens to the surrounding membrane.

The large cavity in our model of TMC1 raises the possibility that the permeation pathway in TMC is larger than has been seen in most ion channel proteins (*Goldschen-Ohm and Chanda, 2017*). In addition, the permeability of hair cells to FM1-43 (~11 Å minimal diameter, MW 0.56 kDa) (*Gale et al., 2001*; *Meyers et al., 2003*), Texas Red-labeled aminoglycosides (~15 Å minimal diameter, MW 1.29–1.43 kDa) (*Alharazneh et al., 2011*; *Marcotti et al., 2005*) and Texas Red-labeled 3 kDa dextran (*Figures 5–9*), suggests the presence of a unusually large permeation pathway. The time course for labeling of hair cells by these fluorescent molecules varies with their size, with the maximal labeling requiring a few minutes for FM1-43 (*Alharazneh et al., 2011*; *Meyers et al., 2003*), 30–60 min for Texas Red-labeled aminoglycosides (*Alharazneh et al., 2011*) and 1–2 hr. for Texas Red-labeled 3 kDa dextran, consistent with these molecules permeating through the MET channel. As far as we are aware, the large conductance mechanosensitive channel (MscL) is the only other ion channel known to be permeable to 3 kDa dextran-TR (*Doerner et al., 2012*). MscL has no ion selectivity and its open pore has been estimated to have a minimal diameter >25 Å (*Perozo et al., 2002*). In the case of MscL, uptake of dextran-TR occurs over minutes (*Doerner et al., 2012*), whereas our most consistent uptake was measured after 2 hr, suggesting that the rate of dextran permeation through the pore of the MET channel is considerably lower than MscL. Although it is conceivable that dextran permeation occurs in a conformation of the MET channel that is only rarely visited, most likely the pore of the MET channel is considerably smaller than that of MscL. The permeation of Texas Red-labeled aminoglycosides and dextrans would require a pore considerably larger than the 12 Å estimate that is based on the relative permeability of the

MET channel to quaternary ammonium ions (*Farris et al., 2004*). If the permeation pathway of the MET channel has some exposure to the surrounding lipid membrane, as is the case for the TMEM16 proteins, it may also exhibit structural plasticity.

The cavity at the protein-lipid interface in our model of TMC also has important implications for understanding the role of membrane lipids in regulating the activity of the MET channel. Phosphatidylinositol-4,5-bisphosphate (PIP$_2$) is known to regulate a diverse array of ion channel proteins (*Hilgemann et al., 2001*; *Suh and Hille, 2008*), including the MET channel (*Effertz et al., 2017*; *Hirono et al., 2004*). In hair cells, the concentration of PIP$_2$ is carefully regulated such that its concentration is high at the location of the MET channel at the tips of stereocilia. In addition, depletion of PIP$_2$ has been reported to alter the gating properties of the MET channel (*Effertz et al., 2017*; *Hirono et al., 2004*). For example, depletion of PIP$_2$ increases the resting open probability of the MET channel, leads to loss of mechanotransduction current and dramatically alters fast adaptation, a process whereby the MET channel closes in the continual presence of a mechanical stimulus. Moreover, depletion of PIP$_2$ has also been shown to influence single channel conductance and both Ca$^{2+}$ permeability and blockade of the channel (*Effertz et al., 2017*), suggesting an intimate connection between this membrane lipid and the process of ion permeation. It has also been proposed that the influence of Ca$^{2+}$ on the resting open probability of the MET channel is due to interactions of the divalent ion with membrane lipids rather than with a Ca$^{2+}$ binding site on proteins in the MET complex (*Peng et al., 2016*). It will be interesting to use our models of TMC1 to investigate structural interactions between proteins in the MET channel complex, to explore the unique properties of its ion permeation pathway and the role of membrane lipids in hair cell mechanotransduction.

## Materials and methods

### Structural template search

To obtain possible structural templates to build a 3D model of mTMC1 we used two strategies. In the first one, the hidden Markov model (HMM) profile of mTMC1 was obtained after a three-iteration sequence scanning against the NCBI uniclust20 sequence database (dated September 11, 2015) with an E-value inclusion threshold of $10^{-3}$ using the HHblits server (*Remmert et al., 2011*). The mTMC1 HHM profile built from the closest fifty homolog sequences identified in HHblits was used as an input in the HHpred server (*Hildebrand et al., 2009*; *Zimmermann et al., 2018*) and scanned against the HMM profiles corresponding to each of the Protein Data Bank (PDB) crystallographic structures (pdb70 database dated September 12, 2015). nhTMEM16 (PDB id: 4WIS) was selected as a suitable template as it presented the highest coverage (residues 151–646), sequence identity (13%) and correspondence between secondary structural elements. In the second strategy, the single sequence of mTMC1 was used as an input into the HHpred server in two different attempts dated November 20, 2017 and January 30, 2018, and nhTMEM16 was also detected as a template. In both cases (dates), the HMM profile of mTMC1 was obtained after scanning the sequence against the uniprot20 database with three iterations, an E-value inclusive threshold of $10^{-3}$ and a minimum sequence coverage of 20%. The subsequent scan of the resulting HMM profile against the PDB_mmCIF70 database identified nhTMEM16 as a possible template in both runs, as well as mTMEM16A in the run dated January 30, 2018 (Ca$^{2+}$-free and Ca$^{2+}$-bound, PDB ID: 5OYB and 5OYG, respectively). Both nhTMEM16 and mTMEM16A showed excellent coverage (residues 151–744, and 140–755, respectively), highest sequence identity of 12% and 11%, respectively, and best correspondence between the secondary structural elements. The initial HMM-based sequence alignment of mTMC1 and mTMEM16A used in the subsequent steps of modeling was that obtained in the run dated January 30, 2018, while that between mTMC1 and nhTMEM16 was that from the run dated November 20, 2017.

Hydrophobicity profiles for the sequences of mTMC1, nhTMEM16 and mTMEM16A were obtained and aligned using the AlignMe web server (*Stamm et al., 2013*, *Stamm et al., 2014*), using the Hessa, White, and von Heijne (HWvH) hydrophobicity scale for alpha-helical proteins (*Hessa et al., 2005*) and a 13-residue long triangular window for smoothing.

## Models of mTMC1 based on nhTMEM16 and mTMEM16A

The alignment of mTMC1 and nhTMEM16 HMM-based sequence profiles was refined by an iterative process guided by conservation scores obtained using the Consurf server (*Ashkenazy et al., 2016*) as a guide to remove gaps within secondary structural elements and position the most conserved residues in the internal regions of the protein. In the final alignment, the percentage of identical residues between nhTMEM16 and mTMC1 was ~13% and was used in the MODELLER package (*Webb and Sali, 2016*), where 2000 modeling iterations were performed. The ten models with highest MODELLER probability distribution function score (molPDF) were selected, and the model with the best ProQM score (*Ray et al., 2010*) and PROCHECK (*Laskowski et al., 1993*) analysis was selected as the ultimate mTMC1 model.

The sequence alignment between nhTMEM16 and mTMEM16A was derived from the structural alignment of the two structures (PDB IDs: 4WIS and 5OYB, respectively) using TM-align (*Zhang and Skolnick, 2005*). This alignment, together with that refined of mTMC1 and nhTMEM16, were used to obtain an initial sequence alignment between mTMC1 and mTMEM16A, using nhTMEM16 as a guiding sequence. The final alignment between mTMC1 and mTMEM16A was obtained after several iterations of refinement using a similar protocol to that described above, with ~8% identical residues, and was used as input in MODELLER to generate a set of 2000 mTMC1 models. The best mTMC1 model using mTMEM16A as template was selected as described above for the nhTMEM16-based models.

The position of each mTMC1 protomer in the dimer was determined after structural superimposition of each protomer onto the corresponding dimeric X-ray structure oriented in the membrane using the OPM server (*Lomize et al., 2012*). The dimeric models were constructed by superimposing the same model onto both protomers of the corresponding crystal structure. The dimer interface, defined as any residue containing an atom within 5 Å of the other protomer, was then refined using MODELLER.

The mTMC1 topology map was generated with Protter (*Omasits et al., 2014*). Electrostatic potentials were calculated using Adaptive Poisson-Boltzmann Solver (APBS) (*Jurrus et al., 2018*) and the PDB2PQR server (*Dolinsky et al., 2004*). All sequence editing was carried out using the graphical platform of Jalview (*Waterhouse et al., 2009*). Protein structures were viewed with the PyMOL molecular graphics system, v2.0.7 (Schrodinger, LLC, NY) and UCSF Chimera v1.12 (*Pettersen et al., 2004*). Final figures were generated in UCSF Chimera using Persistence of Vision Raytracer (POV-Ray) software v3.6.

## mTMC1 conservation analysis

Sequences homologous to the *Mus musculus* TMC1 protein (UniProt ID: Q8R4P5-1) with a significance E-value of $10^{-4}$ and a sequence identity ranging from 35% to 95% were identified through HMMER (*Finn et al., 2011*) by iterative profiling against the UniRef90 dataset. Of the total 437 unique sequence identified, the closest 150 sequences were aligned using the MAFFT (*Katoh et al., 2002*) server to generate the multiple sequence alignment used to calculate the mTMC1 conservation profile with the Consurf server (*Ashkenazy et al., 2016*). The overall conservation scores were color-coordinately mapped onto the mTMC1 homology model shown in *Figure 3E* and *Figure 3— figure supplement 1*.

## Localization of TMC1 mutations

The table containing TMC1 mutations was modified from (*Bakhchane et al., 2015*) to include the latest reported deafness-causing mutations (R604G, R389Q, N199I, S596R, and A428D) and the deafness mutations identified in mice (Y182C and Y449H). Mouse TMC1-8 and TMEM16A-K protein sequences were aligned independently using Jalview (*Waterhouse et al., 2009*) and Clustal Omega (*Sievers et al., 2011*), and the residue conservation score was calculated using Jalview based on the AMAS method (*Livingstone and Barton, 1993*).

## Reagents

Lysine fixable 3 kDa and 10kDa-FITC or Texas Red-labeled dextrans were purchased form Thermo-Fisher scientific Inc (Waltham, MA.). Dextrans were diluted in Hanks' balanced salt solution without $Ca^{2+}$ and $Mg^{2+}$ (HBSS-CFM) at a final concentration of 10 mg/mL, aliquoted in black opaque tubes

and stored at −30℃ until use. As indicated by the manufacturer, the actual molecular weights of dextran molecules in each preparation has a distribution, and contains polymers with molecular weights ranging from 1.5 to 3.0 kDa (including the dye) in the case of 3 kDa dextran-TR, and 50% of the dextran molecules contains a dye, on average.

Hank's balanced salt solution (HBSS), HBSS-CFM, phosphate buffered saline (PBS), Dulbecco's Modified Eagle Medium (DMEM), DMEM media without calcium, and Leibovitz's L-15 medium with GlutaMAX were also purchased from ThermoFisher scientific Inc. Cell impermeant BAPTA tetrapotassium salt (Molecular Probes, ThermoFisher scientific, Inc.) was dissolved at 50 mM in water. A stock solution of 30 mM benzamil hydrochloride hydrate (Sigma-Aldrich, St. Louis, MO) was prepared in DMSO. Amiloride hydrochloride (EMD MILLIPORE, Burlington, MA) was dissolved at 150 mM in DMSO. Aminoglycosides, dihydrostreptomycin sesquisulfate, and neomycin trisulfate salt hydrate (Sigma-Aldrich) were dissolved at 50 mM in water.

## Dextran uptake experiments

Excision of the temporal bones from P6 mice and further cochleae dissection including removal of the semicircular canals and vestibular organs, were performed in Leibovitz's L15 media with surgical forceps under a Linitron Z650HR amplification stereomicroscope equipped with a WF10X eyepiece and an ACE light source. Two incisions were performed on the dissected cochleae free of surrounding tissue; one on the round window and other at the apical cochlear region. At least three cochleae from littermate mice were used for each experimental condition. Cochleae were placed on a corning PYREX 9 depression plate well and incubated for 2 hr at room temperature in the dark with 3 kDa dextran-TR in Leibovitz's L15 media at a final concentration of 2 mg/mL with gentle shaking in an orbital shaker. After incubation, tissue was washed twice with media, once with HBSS, and fixed in 4% paraformaldehyde (PFA, Electron Microscopy Science, Hatfield, PA) for 30 min. Fixed tissue was washed with HBSS to remove PFA and the spiral ligament and the tectorial membrane were both removed to obtain a fixed organ of Corti in HBSS buffer. Samples were then permeabilized in 0.5% Triton X-100 in PBS containing Alexa Fluor-488 phalloidin (ThermoFisher Scientific) at a 1:200 dilution for 30 min to label F-actin. Tissue was washed 2–3 times with HBSS buffer to remove Triton and phalloidin and once with PBS to remove salt. Organ of Corti tissues were finally mounted with Pro-Long Diamond antifade mounting media (ThermoFisher Scientific) on a superfrost plus microscope slide (Fisherbrand, Pittsburgh, PA) and covered with a #1.5 glass coverslips of 0.17 ± 0.02 mm thickness (Warner Instruments, Hamden, CT) for confocal imaging.

3 kDa dextran-FITC, 10 kDa dextran-FITC, and 10 kDa dextran-TR uptake experiments were performed as previously described for 3 kDa dextran-TR. In the experiments performed simultaneously with 3 kDa dextran-TR and 10 kDa dextran-FITC, both dextrans were added together at a final concentration of 2 mg/mL for each. In these experiments, tissues were counterstained with CF405M phalloidin (Biotum, Fremont, CA) at a 1:50 dilution to label F-actin and prepared for imaging as described above.

Dextran uptake experiments at 4℃ were performed in parallel at room temperature with at least 3 cochleae of wild-type littermate mice per condition. Tissue was prepared as described above, but cochleae were placed in two different corning PYREX 9 depression plates. 3 kDa dextran-TR was added in Leibovitz's L15 media at 2 mg/mL and incubated for 2 hr in the dark in two different orbital shakers, one of them located in the lab space (22℃) and the other one in the cold room (4℃). Tissue was then fixed, permeabilized, counterstained with actin and prepared for imaging as described above.

MET channel blockage experiments with BAPTA were performed by incubating excised cochlea in DMEM media without calcium containing BAPTA at a final concentration of 5 mM for 30 min. After BAPTA incubation, 3 kDa dextran-TR was added at a final concentration of 2 mg/mL. Control samples were prepared by incubating cochleae in regular DMEM media in absence of BAPTA. In the MET channel blockage experiments, organ of Corti explants were incubated in media with dihydrostreptomycin (DHS) at 500 µM, benzamil at 30 µM, neomycin at 500 µM or amiloride at 150 µM for 30 min prior to addition of 3 kDa dextran-TR at 2 mg/ml in cell media containing the corresponding MET blocker. A control sample was prepared in the same way but incubated for 30 min in cell media with the corresponding volume of water or DMSO prior to addition of the dextran. 3 kDa dextran-TR was incubated in the presence or absence (control) of blockers for 2 hr at RT. Tissue was washed, fixed and counterstained with Alexa488-phalloidin as previously described for confocal image

acquisition. Stock solutions were prepared for the MET blockers and stored at −70°C until use, and a new aliquot was thawed and used in each experiment.

## Mouse strains and TMC1/TMC2 KOs experiments

TMC1 KO, TMC2 KO mice (RRIDs: IMSR_JAX:019146 and IMSR_JAX:019147) (*Kawashima et al., 2011*), and transgenic mice endogenously expressing TMC1 fused at the C-terminal to Cherry fluorescent protein in a TMC1/TMC2 dKO background (TMC1-cherry, RRID: IMSR_JAX:028392) (*Kurima et al., 2015*) were kindly donated by Dr. Andrew Griffith (NIDCD-NIH). Genomic DNA extraction from tails snips and genotyping PCR reaction were performed using MyTaq Extract-PCR kit (Bioline, Taunton, MA). TMC1 KO mice were genotyped using the primers listed in the table below (*Table 1*) and a separated PCR protocol published in the Jackson Laboratory website (https://www.jax.org, B6.129-Tmc1^tm1.1Ajg/J) and TMC2 KO mice were genotyped as previously described (*Pan et al., 2013*). PCR product was run on a 2% agarose gel and the Quick-load 100pb DNA ladder (New England Biolabs Inc., Ipswich, MA) was used for fragment size visualization. TMC1/TMC2 dKO mice were generated by breeding TMC1 and TMC2 single KOs and the colony was expanded by breeding TMC1/TMC2 dKO mice with transgenic TMC1-cherry mice. P6 mice used in TMC1/TMC2 dKO experiments were genotyped postmortem using the primers for the cherry fusion protein (mCherry) listed in *Table 1*.

Wild-type mice of the same age and strain (C57BL/6J; The Jackson Laboratory Bar Harbor, ME) were used as controls. TMC1 and/or TMC2 heterozygous mice were also used as controls and gave similar results compared to wild-type C57BL/6J mice (data not shown).

## Plasma membrane expression in HEK293T cells

mTMC1, chimera 1, and chimera 2 pcDNA3.1+C myc plasmids were synthesized and codon optimized (Genscript, Piscataway, NJ). Murine TMEM16A isoform ac tagged in the C-terminus with a GFP (TMEM16A-GFP) was generously provided by Criss Hartzell. A tdTomato fluorescent protein containing the calreticulin signal sequence and a KDEL endoplasmic reticulum retention site (tdTomato-ER3, plasmid#58097, kindly donated by Michael Davidson) was purchased from Addgene (Cambridge, MA). TMEM16A-GFP and tdTomato-ER3 were used as positive controls for plasma membrane and endoplasmic reticulum expression, respectively.

One day before transfection, HEK293T cells (RRID: CVCL_0063, ATCC, Manassas, VA) growing in DMEM without calcium supplemented with 10% FBS and GlutaMAX (ThermoFisher scientific, Inc.) were seeded at 30% confluency in six-well plates containing collagen-treated glass coverslips. The next day, HEK293T cells were transiently transfected using FUGENE6 transfection reagent (Promega, Madison, WI) with each one of the 5 cDNAs described above. 18 hr after transfection, cells were washed with PBS and fixed with 2% PFA in PBS for 30 min. Fixed cells were washed with PBS and incubated with 0.5% Triton X-100 in PBS for 30 min to permeabilize the cells. After removing TritonX-100 with 2–3 PBS washes, 4% BSA (Probumin, Millipore, KanKakee, Il) in PBS was added to the cells and incubated for 1 hr. Primary rabbit polyclonal antibody anti-Myc tag antibody (ab9106, RRID: AB_307014, Abcam, Cambridge, United Kingdom) at a 1:500 dilution in PBS with 4% BSA was added and incubated with the cells for 1 hr at RT. After 2–3 washes with PBS to remove unbound antibody, secondary AlexaFluor488-conjugated goat anti-rabbit (ThermoFisher scientific, Inc.) at a 1:2000 dilution in PBS with 4% BSA was added and incubated for 30 min together with CF405M-Phalloidin (Biotium) at a 1:100 dilution to label F-actin. Cells were then washed with PBS several

**Table 1.** Primers used for the genotyping of TMC1 and TMC2 KO and TMC1-Cherry transgenic mice.

| | Forward primer | Reverse primer | PCR frag. |
|---|---|---|---|
| TMC1 wt | GACAGTTGGTGCTGGGATCT | AAGTACGAGGCCACTGAGGA | 215 bp |
| TMC1 KO | ATCCTCTGCATGGTCAGGTC | CGTGGCCTGATTCAT | 315 bp |
| TMC2 wt | CGGTTCTTCTGTGGCATCTTACTT | ACCAGGCAATTGACATGAATA | 401 bp |
| TMC2 KO | CTGCCTTCTGGTTAGATCACTTCA | GTGTTTTAAGTGTACCCACGGTCA | 621 bp |
| mCherry | TTCACTTGCCCTTCTTCATCT | CGCCCTCGATCTCGAACT | 167 bp |

DOI: https://doi.org/10.7554/eLife.38433.029

times and mounted using ProLong Diamond antifade mounting media (ThermoFisher scientific, Inc.) in microscopy slides for confocal imaging.

## Image acquisition

Super-resolution imaging was performed in the Neuroscience Light Imaging Facility (NINDS) with a confocal laser scanning microscope Zeiss LSM 880 (Carl Zeiss AG, Oberkochen, Germany) equipped with a 32 channel Airyscan detector (*Korobchevskaya et al., 2017*). The whole organ of Corti images were taken in SR mode with a 20X objective (Carl Zeiss). To image the hair cells, we collected a z-stack of images from the stereocilia to the apical half of the hair cell body. We used oil immersion alpha Plan-Apochromat 63X/1.4 Oil Corr M27 objective (Carl Zeiss) and Immersol 518F immersion media ($n_e$ = 1.518 (30°), Carl Zeiss). Identical image acquisition settings, no averaging, and optimal parameters for x, y, and z resolution were used in all samples from each independent experiment. Image acquisition and Airyscan image processing were done with Zen Black 2.3 SP1 software (Carl Zeiss) using the Airyscan 3D reconstruction algorithm with the automatic default Wiener filter settings.

Confocal imaging on transfected HEK293 cells was performed in the Microscopy and Imaging Core (NICHD) with an inverted laser scanning microscope Zeiss LSM 780 (Carl Zeiss) equipped with a motorized stage, definite focus and a high sensitivity GaAsp multi-channel spectral detector. We used a 63X/1.4 objective Plan-Apochromat (Carl Zeiss) and the Zen software (Carl Zeiss) for image acquisition.

## Image data processing and fluorescence quantification

Microscopy data processing, analysis, and quantification were done in ImageJ (*Schneider et al., 2012*). To measure the fluorescence intensity on the hair cell body, we generated a region of interest (ROI) inside the cell body using the oval tool and measured the mean fluorescence intensity of each ROI. The mean fluorescence intensity of an equivalent ROI in an area outside the hair cells was considered as background and subtracted from the values at the cell body for each image.

To quantify the fluorescence intensity at the different cochlear regions, we measured the mean fluorescence intensity at the hair cell body in three images at each cochlea region (basal, middle, and apical), considering a total of 70 OHC and 19 IHC form the basal, 52 OHC and 17 IHC from the middle, and 59 OHC and 15 IHC from the apical region. Four independent images of the cochlear middle region were used to quantify the mean fluorescence intensity of a total of 136 HC at RT and 140 HC at 4°C. In the MET blocker experiments, we measured the mean fluorescence of 3–5 images for the middle cochlea region resulting in a total of 133, 116, 72, 111, 64 and 80 HC in the control, BAPTA, amiloride, benzamil, DHS, and neomycin condition, respectively. In the TMC KOs experiments, mean fluorescence was measured for 4 images of the middle cochlear region for each mouse strain. The number of cells measured was 146 for wild-type, 91 for TMC1 KO, 111 for TMC2 KO and 94 for TMC1/TMC2 dKO. Data was initially processed in Microsoft Excel, and GraphPad Prism V.7 software (GraphPad software, La Jolla, CA) was used to generate the graphs and perform the statistical analysis. Representative confocal images for each condition in a representative experiment are shown with the same display range. Background subtraction was performed using a rolling ball of 200 pixels for the figures shown in the manuscript.

3D volume reconstruction of 3 kDa dextran-TR fluorescence was generated using the Volume Viewer 2.01 plugin contained in the ImageJ software (*Schneider et al., 2012*). A series of images along approximately 8 µm on the z-axes was collected from an IHC with Airyscan super-resolution. Airyscan image processing was performed with Zen software using the default settings. The image was cropped using ImageJ to show one single cell and dextran channel was further processed with the volume Viewer 2.01 plugin.

## Acknowledgments

We thank Anirban Banerjee, Alex Chesler, Andy Griffith, Lucy Forrest, Katie Kindt, Mark Mayer, Joseph Mindell and members of the Swartz laboratories for helpful discussions. We thank Lucy Forrest for feedback on the TMC1 homology model generation and computational analysis, Vincent Schram from the National Institute of Child Health and Development microscopy and imaging core for assisting in the confocal image acquisition, Caroline Smith from the National Institute of

Neurological Disorders and Stroke (NINDS) light imaging facility, Tsg-Hui Chang for invaluable help with colony management and mice care, H. Criss Hartzell from the Emory University School of Medicine for sharing the mTMEM1A-GFP cDNA and Michael Davidson for sharing the tdTomato-ER3 cDNA with the scientific community through Addgene. We also thank Andrew J. Griffith from the National Institute on Deafness and other Communication Disorders (NIDCD) for generously providing TMC1 KO, TMC2 KO, and TMC1-cherry transgenic mice strains, and Kiyoto Kurima and Soumya Korrapati (NIDCD), as well as the NIDCD sequencing facility for assisting us with the TMC mice genotyping. This research was supported by the Intramural Research Program of the NINDS, NIH, Bethesda, MD to KJS. A.B. was supported by the Intramural Research Program of the NINDS, NIH, and by a Robert Wenthold Postdoctoral Fellowship from the Intramural Research Program of the NIDCD. C.F-F was supported by the Intramural Research Programs of the NINDS, the National Institute of Mental Health and by the NIDCD, NIH, Bethesda, MD.

## Additional information

### Competing interests

Kenton Jon Swartz: Reviewing editor, *eLife*. The other authors declare that no competing interests exist.

### Funding

| Funder | Grant reference number | Author |
| --- | --- | --- |
| National Institute on Deafness and Other Communication Disorders | Robert Wenthold Postdoctoral Fellowship | Angela Ballesteros |
| National Institute of Neurological Disorders and Stroke | Intramural Research Program NS002945 | Angela Ballesteros Cristina Fenollar-Ferrer Kenton Jon Swartz |
| National Institute of Mental Health | Intramural Research Program | Cristina Fenollar-Ferrer |
| National Institute on Deafness and Other Communication Disorders | Intramural Research Program | Cristina Fenollar-Ferrer |

The funders had no role in study design, data collection and interpretation, or the decision to submit the work for publication.

### Author contributions

Angela Ballesteros, Conceptualization, Data curation, Formal analysis, Validation, Investigation, Methodology, Writing—original draft, Writing—review and editing; Cristina Fenollar-Ferrer, Conceptualization, Data curation, Formal analysis, Validation, Investigation, Visualization, Writing—review and editing; Kenton Jon Swartz, Conceptualization, Resources, Supervision, Funding acquisition, Writing—original draft, Project administration, Writing—review and editing

### Author ORCIDs

Angela Ballesteros (iD) http://orcid.org/0000-0001-5796-6875
Kenton Jon Swartz (iD) http://orcid.org/0000-0003-3419-0765

### Ethics

Animal experimentation: The animal care and experimental procedures were performed in accordance with the Guide for the Care and Use of Laboratory Animals and were approved by the Animal Care and Use Committee of the National Institute of Neurological Disorders and Stroke (Animal protocol number 1336).

**Decision letter and Author response**
Decision letter https://doi.org/10.7554/eLife.38433.038
Author response https://doi.org/10.7554/eLife.38433.039

## Additional files

**Supplementary files**
• Transparent reporting form
DOI: https://doi.org/10.7554/eLife.38433.030

**Data availability**
The PDB files of the TMC1 models have been provided as Figure 2-source data 1. Source data files have also been provided for Figures 4, 5 and 7-9.

The following previously published datasets were used:

| Author(s) | Year | Dataset title | Dataset URL | Database, license, and accessibility information |
|---|---|---|---|---|
| Dutzler R, Brunner JD, Lim NK, Schenck S | 2014 | Crystal structure of the lipid scramblase nhTMEM16 in crystal form 1 | https://www.rcsb.org/structure/4WIS | 4WIS |
| Paulino C, Kalienkova V, Lam KM, Neldner Y, Dutzler R | 2017 | Structure of calcium-bound mTMEM16A chloride channel at 3.75 A resolution | https://www.rcsb.org/structure/5OYB | 5OYB |
| Paulino C, Kalienkova V, Lam KM, Neldner Y, Dutzler R | 2017 | Structure of calcium-free mTMEM16A chloride channel at 4.06 A resolution | https://www.rcsb.org/structure/5OYG | 5OYG |

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
