## [Decision Letter]

[Editors’ note: this article was originally rejected after discussions between the reviewers, but the authors appealed against the decision and a revised paper was accepted.]

Thank you for submitting your work entitled "Structural relationship between the putative hair cell mechanotransduction channel TMC1 and TMEM16 proteins" for consideration by *eLife*. Your article has been reviewed by three peer reviewers, including Baron Chanda as the Reviewing Editor and Reviewer #1, and the evaluation has been overseen by a Senior Editor.

Our decision has been reached after consultation between the reviewers. Based on these discussions and the individual reviews below, we regret to inform you that your work will not be considered further for publication in *eLife*.

In this study, the authors develop a structural model of TMC 1 based on its homology to the TMEM16 proteins. The TMC1 is a multi-transmembrane protein which is thought to be an essential pore-forming subunit of hair cell mechanotransduction complex. Direct evidence that the TMC1 is a channel is lacking because these channels are not expressed in any heterologous system. Thus, despite tremendous interest, these proteins remain recalcitrant to structural and functional analysis. This study gets around these hurdles by building a high-quality homology model of the channel and testing the predictions of this model by sizing the pore of native channels in hair cells. While the reviewers agree that the overall approach is creative, they are not convinced that the sizing experiments with variously sized dextrans are sufficiently specific to discriminate between the authors' model and other existing models. Attached below are the individual reviews, some of which have been amended to reflect the full scope of discussion between the reviewers.

Reviewer #1:

The vertebrate mechanotransduction channel in inner hair cells converts the mechanical stimulus produced as a result of sound waves into electrical signals. Although the molecular identity of this mechanotransduction channel complex has not been established, recent studies suggest that transmembrane channel-like 1 and 2 proteins are part of the pore-forming subunit of this complex. Bioinformatic studies have previously suggested that the TMC proteins are phylogenetically related to the TMEM16 ion channels, which are a family of calcium-activated chloride channels. In this study, Ballesteros et al. exploited the newly available high-resolution structures of TMEM16 ion channel to build a homology model of the TMC 1 protein. TMEM 16 proteins are known to act as lipid scramblases and therefore it is not surprising that the structural model predicts a large lipid-facing cavity. Unfortunately, it is difficult to test this model directly because TMC1 proteins do not traffic to the cell membrane. A creative attempt to overcome this limitation by creating fusions between TMEM16 and TMC1 also did not result in functional expression. The authors then tested their model by measuring the rate of uptake of fluorescently labeled dextrans in native hair cells. Their studies reveal that the 3KD but not 10 KD dextrans are able to permeate through these channels allowing them to size the permeation pathway.

This is a timely study and a detailed structural model of the TMC1 has the potential to guide the next set of experiments. However, it is not clear to this reviewer whether the structural link between TMC1 protein and TMEM16 has been established in this study. Based on their modeling, the authors predict that this channel has more transmembrane segments than predicted by others, but this prediction was not specifically tested. Given the challenges of working with this system, I am not sure if this limitation of this study can be overcome. The Dextran experiments are interesting and shed new information about the pore of the mechanotransduction channel, but they do not conclusively establish that the TMC1 protein is structurally related to TMEM16. I wonder if the authors can use more specific tests such as measuring the lipid scramblase activity to establish a more specific functional link between TMC1 and TMEM16.

Reviewer #2:

This interesting and thought provoking manuscript Ballestreros and colleagues investigate the structural relationship between the TMC1 channel and the TMEM16 proteins. The TMC1 channel plays a key role in mechanotransduction, allowing these cells to transform sound waves in electrical signals. While the role of this channel in physiology is well established, our mechanistic understanding is more limited since attempts to heterologously express it have been unsuccessful so far. The authors build a homology model of the channel based on its reported similarity to the nhTMEM16 scramblase and to the TMEM16A channel. This model suggests that this protein forms a wide, membrane-exposed hydrophilic cavity, which they propose serves as the ion permeation pathway. To test this hypothesis the authors show that hair cells are able to uptake large molecules, such as 3 kDa dextrans, in a manner that is dependent upon the activity of the TMC1 and 2 channels.

Overall, this is an interesting manuscript that addresses an important question. Their structural model is intriguing and thought provoking. However, while the manuscript has merits, the intrinsic limitations of the experimental system prevent the authors from rigorously testing their proposed model. The dextran permeation experiments are evocative, but not a definitive proof of their model. The authors cannot rule out that permeation occurs via other channels in the hair cells; heterologous expression of TMEM16A/TMC1 chimeras fail to traffic to the plasma membrane preventing structure-function studies or experiments aimed at determining the topology of the channel.

In addition to these concerns, I have a few concerns:

1) TMC1 is highly cation selective and discriminates between mono- and divalent cations. Such a selectivity profile is difficult to reconcile with the hypothesis that these ions are permeating through a pathway that is wide enough to accommodate 3kDa dextrans and is partly lipid lined. Indeed, the TMEM16 scramblases that also function as ion channels are poorly selective and do not discriminate between anions and cations. I wonder if one of the two models built using the TMEM16A channel serve as a better template to think of TMC1 as an ion channel. Indeed, the dextran uptake observed here is slow. This could suggest that TMC1 visits a dextran-permissive conformation only rarely, or that it allows dextran to go through very slowly. This should be discussed.

2) A second, and related, point is that most of the human deafness missense mutations cluster to the periphery of the pathway, and only one (at least only one that I could see in Figure 4) inside the pathway. The authors could analyze their model 2 (based on the Ca^2+^-bound TMEM16A structure) concerning the placement of the human mutations and see if a higher fraction would face the inside of the pore. This possibility should be discussed.

3) The authors should compare their models to previously published ones, based on other proposed topologies and score them quantitatively.

4) I am somewhat confused by the data in Figure 4—figure supplement 2. In cells expressing TMEM16A and the TMEM16A/TMC1 chimeras the distribution of F-actin changes quite dramatically. For example, in the top panel there is a single sharp peak corresponding to the plasma membrane, whereas in the third panel from the top there are two peaks. Does this mean that the authors are measuring across multiple plasma membranes? The placement of the white reference bar would suggest otherwise. More generally, the F-actin distribution in all panels is significantly broader compared to the top panel. What does this imply?

Reviewer #3:

The multi-transmembrane protein TMC1 is important for hair cell mechanotransduction and has been proposed to be the pore-forming subunit of the channel. However, direct evidence is missing that TMC1 is an ion channel because it has not been possible to express TMC1 at the surface of cells to characterize its properties. While TMC1 can be localized to stereocilia of hair cells, it remains in the ER in heterologous cells. The authors now generated structural models of TMC1 based on structural information available for the TMEM16, a calcium activated chloride channel. While there is no substantial sequence homology between the two proteins, the authors argue that their predicted hydrophobicity profile is similar thus justifying homology modelling. The models predict that TMC1 has 10 transmembrane domains and a relative large cavity that could serve as an ion permeation pathway. The authors also show that hair cells can take up 3 kDa dextran and that this uptake is blocked when mechanotransduction channels are blocked or in mice with mutations in TMC1/2. The authors take this as support for their structural model, which predicts a relatively large pore for the channel.

While this paper begins to investigate intriguing similarities between TMC and TMEM16 proteins, the findings remain speculative. It has also been suggested before that TMC proteins might be channels with a relatively large permeation pathway. Others have already suggested that TMC proteins might have similarities to TMEM16 (Hahn, 2009; Medrano-Soto, 2018). The experimental data in the manuscript are not directly related to the structural predictions.

Specific comments:

1) The model of the authors is low-resolution and predicts a 10 TM domain topography, which is different from the reported 6TM domain structure that was based on experimental data (Labay and Griffith, 2010) or the 8 TM structure that has been proposed by others. Without experimental validation, the 10TM domain remains an interesting idea but has now strong experimental foundation. Membrane topography could be mapped by expressing epitope tagged molecules in hair cells. This is admittedly a very difficult experiment but ultimately necessary to establish protein topography (or to achieve expression in a heterologous system).

2) Efforts by the authors to experimentally verify the model have unfortunately not met with success. They generated chimera between TMC1 and TMEM16 that remained in the ER. Thus, the paper is lacking sufficient experimental context to validate the significance of the proposed structural similarities between TMEM16 and TMC proteins.

3) The observation that 3 kDa dextran may enter hair cells through the transduction channel does not establish a structural link between TMC1 and TMEM16. It is already established that the pore of the hair cell mechanotransduction channel is large, and that permeation through the pore depends upon TMC1/2 proteins and intact tip links. A variety of large compounds have been shown to permeate the hair cells transduction channel. Far more detailed studies on the uptake and effects of conditions that change channel properties would be important to investigate dextran uptake (e.g. how is uptake affected by changes in internal and external Ca^2+^ that affect gating properties of the transduction channel) (see Farris and Ricci, 2014). Even if these experiments are performed, it is hard to link the results to a particular molecule because the compound does not have specific characteristics that would tell about the specific protein that lines the pore.

4) There is some concern that the data do not directly show that the dextran compound penetrates through the channel. Uptake takes 2 hours, which is a very slow time frame. Other large compounds that go through the channel such as FM143 enter in 10-20 seconds. The cells take up more compound during an extended time period through endocytosis. Breaking transduction channels or tip links does not directly address whether the compound goes through the channel. Studies from the Fettiplace lab have shown that when you inactivate transduction channels, a "reverse polarity channel" become activated in hair cells, which could also be permeant for the dye.

---

## [Author Response]

[Editors’ note: the author responses to the first round of peer review follow.]

Our decision has been reached after consultation between the reviewers. Based on these discussions and the individual reviews below, we regret to inform you that your work will not be considered further for publication in eLife.

Thank you for overseeing the review of our recent submission to *eLife* on the structural relationship between TMC and TMEM16 proteins. We have carefully gone through the reviews comments and would like to appeal the decision. We appreciate that the focus of our work lies at the interface between different communities and that it may not be straightforward for everyone to appreciate the contribution of our study. In the following we respond to each of the points raised by reviewers, and outline areas where we have revised the manuscript to address the reviewer’s comments.

In this study, the authors develop a structural model of TMC 1 based on its homology to the TMEM16 proteins. The TMC1 is a multi-transmembrane protein which is thought to be an essential pore-forming subunit of hair cell mechanotransduction complex. Direct evidence that the TMC1 is a channel is lacking because these channels are not expressed in any heterologous system. Thus, despite tremendous interest, these proteins remain recalcitrant to structural and functional analysis. This study gets around these hurdles by building a high-quality homology model of the channel and testing the predictions of this model by sizing the pore of native channels in hair cells. While the reviewers agree that the overall approach is creative, they are not convinced that the sizing experiments with variously sized dextrans are sufficiently specific to discriminate between the authors' model and other existing models. Attached below are the individual reviews, some of which have been amended to reflect the full scope of discussion between the reviewers.

We believe the core reason for rejecting our manuscript is based on a misunderstanding of what our modelling provides. This work started in our lab over three years ago when we undertook an HHpred search of structural databases to see if any proteins of known structure exist that are related to the TMC proteins. The E-values, sequence coverage, and other results obtained from those HHpred searches shown in Figure 1—figure supplement 1 reveal that the TMC proteins are highly likely to be related to TMEM16 proteins. We then analyzed the hydrophobicity profiles (Figure 1), which clearly show that the membrane spanning architecture of TMC matches that of the known structure of the TMEM16 proteins. The purpose of the model was to further interrogate the relationship to see if homology models of TMC based on the three available TMEM16 structures could be built that could then withstand structural interrogation using ProQM, a powerful computational tool used for evaluating the quality of structural membrane protein models. That analysis shown in Figure 2 and Figure 2—figure supplement 2 reveals that the three models score almost as high as the structures on which they are built, providing additional confidence in the prediction that the fold of the TMC and TMEM16 proteins are remarkably similar throughout the entirely of the transmembrane region and some N- and C- terminal helices. Although there are many specific features of our models that would be exciting to experimentally test if that were possible (Is the cavity open to the membrane? Do other subunits contribute to the permeation pathway?), that the folds of TMC and TMEM16 proteins are similar is a solid conclusion of our work. We also note that the only two autosomal dominant hearing loss mutations identified to date are located inside the cavity in our models (see below), suggesting that the cavity plays an important role in TMC function. Moreover, electrophysiology data showing that one of these mutations affects MET channel single channel conductance suggests that this residue contributes to forming the permeation pathway and agreeing with its localization in our model.

We were surprised that the reviewers have requested a detailed comparison between our model and the previously published topology models of TMC proteins. From reading the comments we wonder whether there is confusion about the nature of those previous models. They are simply models for the transmembrane topology of TMC proteins (not an attempt to deduce the fold, as is the case in our model) that are based largely on transmembrane prediction programs and having been published as speculation, or in one case accompanied by biochemical experiments on proteins that were retained in the ER. Although we are concerned about the extent to which ER-retained proteins may be misfolded, that data are largely consistent with our model, it’s just that they are also consistent with many other models too.

We would also like to comment on the two related publications proposing a relationship between TMC and TMEM16 proteins. The first published in 2009 shows a partial sequence alignment that suggests the proteins may be related. The second was published in March of 2018 while we were preparing our manuscript for submission to *eLife*. That work uses bioinformatic tools to provide further support for a relationship between the TMC and TMEM16 proteins, but the focus of that work is characterizing the TMEM16 superfamily and they do not computationally explore the relationship to derive strong conclusions. We openly cite these studies in our introduction and give the authors credit for providing evidence of a relationship, but we feel that our work is more extensive and allows much stronger conclusions to be drawn. We would also hope that the *eLife* no-scoop policy would influence the extent to which the most recent publication could be viewed as lowering the novelty or impact of our work.

Establishing that the fold of TMC and TMEM16 proteins are very similar is important because it helps to explain a range of observations that have not been previously discussed in a structural context, including the very large single channel conductance of the MET channel, its permeability to relatively large molecules, including Texas Red-labeled aminoglycosides, and to the influence of PIP2 on both the gating and permeation properties of the MET channel. In the second part of our manuscript we tested whether even larger fluorescently-labeled dextrans could permeate hair cells, as might be expected based on the unusual predicted permeation pathway. The extensive results shown in Figure 5, Figure 6, Figure 7, Figure 8 and Figure 9 (with supplements) clearly establishes that 3kDa Texas Red dextrans can permeate hair cells and that this requires TMC1/TMC2 proteins as well as functional MET channels. We appreciate that these experiments are necessarily indirect and do not allow us to conclude that dextrans directly permeate the TMC proteins. However, our results rule out endocytosis as the mechanism for the diffuse uptake that we see, and we have used cautious language throughout the manuscript to reflect the indirect nature of our permeation experiments.

Collectively, we think that our work establishing a relationship between TMC and TMEM16 proteins is a significant breakthrough for the MET channel field and that it will provide the stimulus for a lot of important work in other laboratories.

Reviewer #1:[…] This is a timely study and a detailed structural model of the TMC1 has the potential to guide the next set of experiments. However, it is not clear to this reviewer whether the structural link between TMC1 protein and TMEM16 has been established in this study. Based on their modeling, the authors predict that this channel has more transmembrane segments than predicted by others, but this prediction was not specifically tested. Given the challenges of working with this system, I am not sure if this limitation of this study can be overcome. The Dextran experiments are interesting and shed new information about the pore of the mechanotransduction channel, but they do not conclusively establish that the TMC1 protein is structurally related to TMEM16. I wonder if the authors can use more specific tests such as measuring the lipid scramblase activity to establish a more specific functional link between TMC1 and TMEM16.

Please see the discussion above.

Reviewer #2:[…] Overall, this is an interesting manuscript that addresses an important question. Their structural model is intriguing and thought provoking. However, while the manuscript has merits, the intrinsic limitations of the experimental system prevent the authors from rigorously testing their proposed model. The dextran permeation experiments are evocative, but not a definitive proof of their model. The authors cannot rule out that permeation occurs via other channels in the hair cells; heterologous expression of TMEM16A/TMC1 chimeras fail to traffic to the plasma membrane preventing structure-function studies or experiments aimed at determining the topology of the channel.

Please see the above discussion.

In addition to these concerns, I have a few concerns:1) TMC1 is highly cation selective, and discriminates between mono- and divalent cations. Such a selectivity profile is difficult to reconcile with the hypothesis that these ions are permeating through a pathway that is wide enough to accommodate 3kDa dextrans and is partly lipid lined. Indeed, the TMEM16 scramblases that also function as ion channels are poorly selective and do not discriminate between anions and cations. I wonder if one of the two models built using the TMEM16A channel serve as a better template to think of TMC1 as an ion channel. Indeed, the dextran uptake observed here is slow. This could suggest that TMC1 visits a dextran-permissive conformation only rarely, or that it allows dextran to go through very slowly. This should be discussed.

These are all great points. Although we think that establishing a structural relationship between TMC and TMEM16 protein is an important step forward, we are at an early stage of understanding structural interactions between proteins that may contribute to forming the MET channel. We discussed the possibility that other proteins or possibly lipids may contribute to forming the permeation pathway, which could also help to generate cation selectivity, but we agree that cation selectivity and a large permeation pathway are seemingly at odds. We hadn’t considered the possibility that the permeability of the MET channel to dextrans might be through a rarely visited conformation, and we now comment on this possibility in the Discussion section.

2) A second, and related, point is that most of the human deafness missense mutations cluster to the periphery of the pathway, and only one (at least only one that I could see in Figure 4) inside the pathway. The authors could analyze their model 2 (based on the Ca^2+^-bound TMEM16A structure) concerning the placement of the human mutations and see if a higher fraction would face the inside of the pore. This possibility should be discussed.

This is an interesting point. The two autosomal dominant deafness-causing mutations (M412 and D569) are located within the cavity in model 1. We apologize if the figure wasn’t clear, but M412 is located near the edge of the cavity, whereas D569 is located deep within the cavity. As suggested, we have also looked at their location in model 2 and model 3 and included these data as Figure 4—figure supplement 1 in the manuscript. Even with movement of the TM 3-6 helices, the two residues remain in similar positions.

3) The authors should compare their models to previously published ones, based on other proposed topologies and score them quantitatively.

As discussed above, the previous models are topology models and there is little to compare quantitatively.

4) I am somewhat confused by the data in Figure 4—figure supplement 2. In cells expressing TMEM16A and the TMEM16A/TMC1 chimeras the distribution of F-actin changes quite dramatically. For example, in the top panel there is a single sharp peak corresponding to the plasma membrane, whereas in the third panel from the top there are two peaks. Does this mean that the authors are measuring across multiple plasma membranes? The placement of the white reference bar would suggest otherwise. More generally, the F-actin distribution in all panels is significantly broader compared to the top panel. What does this imply?

There may be something interesting here, but the point of these figures is to show that TMC does not traffic to the plasma membrane. The different distribution of F-actin observed with the different cDNAs is probably due to a variable distribution of F-actin and presence of actin filaments in the different cells. HEK293 cells are extremely heterogeneous in size and structure, making it difficult to draw any conclusions about potential changes in the actin cytoskeleton.

Reviewer #3:The multi-transmembrane protein TMC1 is important for hair cell mechanotransduction and has been proposed to be the pore-forming subunit of the channel. However, direct evidence is missing that TMC1 is an ion channel because it has not been possible to express TMC1 at the surface of cells to characterize its properties. While TMC1 can be localized to stereocilia of hair cells, it remains in the ER in heterologous cells. The authors now generated structural models of TMC1 based on structural information available for the TMEM16, a calcium activated chloride channel. While there is no substantial sequence homology between the two proteins, the authors argue that their predicted hydrophobicity profile is similar thus justifying homology modelling. The models predict that TMC1 has 10 transmembrane domains and a relative large cavity that could serve as an ion permeation pathway. The authors also show that hair cells can take up 3 kDa dextran and that this uptake is blocked when mechanotransduction channels are blocked or in mice with mutations in TMC1/2. The authors take this as support for their structural model, which predicts a relatively large pore for the channel.

Please see the discussion above that addresses the purpose of our modeling and the weight of our core conclusions concerning the structural relationship between TMC1 and TMEM16 proteins.

While this paper begins to investigate intriguing similarities between TMC and TMEM16 proteins, the findings remain speculative. It has also been suggested before that TMC proteins might be channels with a relatively large permeation pathway. Others have already suggested that TMC proteins might have similarities to TMEM16 (Hahn, 2009; Medrano-Soto, 2018). The experimental data in the manuscript are not directly related to the structural predictions.

Please see the discussion above.

1) The model of the authors is low-resolution and predicts a 10 TM domain topography, which is different from the reported 6TM domain structure that was based on experimental data (Labay and Griffith, 2010) or the 8 TM structure that has been proposed by others. Without experimental validation, the 10TM domain remains an interesting idea but has now strong experimental foundation. Membrane topography could be mapped by expressing epitope tagged molecules in hair cells. This is admittedly a very difficult experiment but ultimately necessary to establish protein topography (or to achieve expression in a heterologous system).

As discussed above, our computational analyses establish that TMC and TMEM16 proteins share a common fold. There would be many aspects of the model we would love to explore experimentally, but we feel that undertaking transfections into hair cells to test the fold would not be a wise use of resources. We also believe that the results in the Labay and Griffith paper are largely consistent with our model. Briefly, their results localizing HA epitopes to the cytosol or ER lumen agree with our model for five of the seven positions studied. One HA epitope tag was inserted into TM4 and would be expected to cause radical structural distortions. Only one of their epitopes (between TM5 and TM6) gave results that are seemingly inconsistent with our model. Their data would suggest that this epitope is cytosolic and our model predicts that it is extracellular. Given concerns with incomplete folding for proteins retained in the ER, we think that their data can be reasonably interpreted as supporting our fold for TMC proteins.

2) Efforts by the authors to experimentally verify the model have unfortunately not met with success. They generated chimera between TMC1 and TMEM16 that remained in the ER. Thus, the paper is lacking sufficient experimental context to validate the significance of the proposed structural similarities between TMEM16 and TMC proteins.

Please see the discussion above.

3) The observation that 3 kDa dextran may enter hair cells through the transduction channel does not establish a structural link between TMC1 and TMEM16. It is already established that the pore of the hair cell mechanotransduction channel is large, and that permeation through the pore depends upon TMC1/2 proteins and intact tip links. A variety of large compounds have been shown to permeate the hair cells transduction channel. Far more detailed studies on the uptake and effects of conditions that change channel properties would be important to investigate dextran uptake (eg. how is uptake affected by changes in internal and external Ca^2+^ that affect gating properties of the transduction channel) (see Farris and Ricci, 2014). Even if these experiments are performed, it is hard to link the results to a particular molecule because the compound does not have specific characteristics that would tell about the specific protein that lines the pore.

We agree that we cannot conclude that dextrans directly permeate the MET channel, but we would point out that we have undertaken a large number of challenging and high-resolution imaging experiments that demonstrate permeation requires TMC1/2 and functional MET channels, and does not involve endocytosis. As the reviewer states, the suggested experiments would not definitively settle the issue. We believe that the results we present make an important contribution and advance the field, and that they have been presented in a clear and cautious manner.

4) There is some concern that the data do not directly show that the dextran compound penetrates through the channel. Uptake takes 2 hours, which is a very slow time frame. Other large compounds that go through the channel such as FM143 enter in 10-20 seconds. The cells take up more compound during an extended time period through endocytosis. Breaking transduction channels or tip links does not directly address whether the compound goes through the channel. Studies from the Fettiplace lab have shown that when you inactivate transduction channels, a "reverse polarity channel" become activated in hair cells, which could also be permeant for the dye.

We agree that it is easier to accept that FM1-43 permeates the MET channel directly because its uptake is relatively rapid. However, as we point out in the discussion, with think the kinetics of uptake for FM1-43, TR-aminoglycosides and TR-dextran vary as would be expected based on their size if all three permeate the MET channel. To provide more information on the kinetics of TR-dextran uptake, we have examined the time course more carefully and show uptake of dextran as early as 60 minutes. We have included an additional Figure5—figure supplement 1 containing this information.

We found this reviewer’s comments about the reverse polarity current seen in the TMC1/2 dKO confusing. The calcium selective Piezo2 channel has been suggested to give rise to this reverse polarity current (Beurg and Fettiplace, 2017), but Piezo has a very small pore and could not conceivably be involved in dextran uptake. Moreover, dextran uptake is ablated under conditions where the reverse polarity current is seen.

The comment about endocytosis is also confusing. We provide direct evidence that endocytosis is not responsible for the diffuse uptake observed with 3kDa dextran. We also provide clear evidence that both 3kDa and 10kDa dextrans can be taken up into small vesicle like structures, clearly showing that we can distinguish endocytic events from the more diffuse pattern of uptake only seen with the 3kDa dextran that requires functional MET channels and TMC1/2 proteins.